# The sexual experience of Italian adults during the COVID-19 lockdown

**Stefano Federici**[1,2]*, **Alessandro Lepri**[1], **Alessandra Castellani Mencarelli**[1,2], **Evel Zingone**[3], **Rosella De Leonibus**[3], **Anna Maria Acocella**[3], **Adriana Giammaria**[4]

**1** Department of Philosophy, Social & Human Sciences and Education, University of Perugia, Perugia, Italy, **2** Myèsis, Research and Development Company, Rome, Italy, **3** Expressive Gestalt Psychotherapy Institute of Perugia, Perugia, Italy, **4** Independent Researcher, Udine, Italy

\* stefano.federici@unipg.it

## Abstract

From March 11 to April 26, 2020, the Italian government imposed a nationwide COVID-19 lockdown, a quarantine that resulted in significant restrictions on the movement and social contacts of the population, with a view to limiting the pandemic outbreak. The quarantine forced people to experience distorted social distance in two contrasting ways. For some people, it resulted in social distancing and isolation, for example by separating noncohabiting couples into different dwellings. For others, however, quarantine increased and imposed social closeness, forcing couples and families into constant, daily, and prolonged cohabitation. The aim of this study was to investigate the sexual health and behaviors of Italian adults during the lockdown period using a multimethod research. An open- and a closed-ended e-questionnaire were administered immediately after the end of the lockdown. A total of 465 Italian adults completed the digital questionnaire (female = 78.7%). Participants recognized their lived sexual experience with generally positive characteristics (related to openness, unproblematic relationship with the body, and awareness and self-reflection about one's sexuality), while negative thoughts such as worry and pain were quite scarce. Participants with a disability (5.6%) showed a marked inversion compared to the mean of respondents, recognizing themselves mainly in negative thoughts related to low self-esteem, inadequacy, and feelings of suffering, yet reporting a higher than mean level of arousal. In the qualitative analysis, the Frequency-Inverse Document Frequency (TF-IDF) index was computed to measure the salience of the word used by participants to respond to the open-ended five questions. It revealed a generally depressed emotional experience associated with the experience of lockdown, both in terms of desire, which seemed to be shifted more to the level of imagination and fantasies, and the actual possibility of experiencing sexual activity as usual. Nevertheless, the participants emphasized an opening to new possibilities in terms of expressing sexuality, accompanied by a rediscovery of the value of tenderness and affectivity as well as a clearer awareness of their sexual life, needs, and desires.

**Data Availability Statement:** All raw data are available from the following database: Federici, S., Lepri, A., Castellani Mencarelli, A., Zingone, E., De Leonibus, R., Acocella, A. M., & Giammaria, A.

(2022). Raw data about the sexual experience of Italian adults during the COVID-19 lockdown. Researchgate.net. https://doi.org/10.13140/RG.2.2.13355.36640.

**Funding:** The authors received no specific funding for this work.

## Introduction

COVID-19 (coronavirus disease 2019; the pathogen called SARS-CoV-2; previously 2019-nCoV) is an acute and highly contagious viral disease [1] that affects the respiratory system. It was first detected in Wuhan, China, at the end of December 2019 [2]. Italy was the first European country to announce an outbreak of the infection on February 21, 2020; the virus rapidly affected the north of Italy and then spread to all other regions [3]. The epidemic caused by COVID-19 has made it necessary to implement strategies to contain the contagion, ranging from the so-called "quarantine," which consists in the restriction of activities or separation of subjects who may have been exposed to the pathogen, to isolation, i.e. the separation of subjects who have contracted the disease from healthy subjects, in order to avoid contagion and further spread of the virus [4, WHO, 5]. From March 11 to April 26, 2020, the Italian government imposed a nationwide "lockdown"—a quarantine that resulted in significant restrictions on the movement and social contacts of the population. These measures, as well as other interventions of a medical-healthcare nature, although necessary, have involved neither a few nor negligible "side effects."

In a scientific article published online in the prestigious British medical journal The Lancet on February 26, 2020—when Italy had already exceeded 400 infected and counted the first deaths from COVID-19 and the first cases were appearing in other European countries—a team of researchers in medical psychology led by the psychologist G. James Rubin [4], from King's College London, reviewed 24 studies that had investigated the psychological impact of quarantine following contagious diseases in recent years (Ebola, SARS, equine influenza). The main negative effects reported by those studies were anger, irritability, sense of confusion, and symptoms of post-traumatic stress disorder. In addition, a literature review by Chew et al. [6] evidenced fears, anxieties, and depression as common psychological symptoms reported across outbreaks; frustration and boredom over the uncertainty of the situation were also found to be present [7]. The various stressors included fear of contagion and loved ones' well-being, the scarce or insufficient availability of products considered essential for a good daily life (for example, a working cell phone and an Internet connection that allows communication on social networks), disruptions in daily routine and work life, the treatment process, and information pertaining to the disease [4, 6]. Moreover, feelings of isolation, abandonment, and stigmatization may have been experienced by survivors of the disease and by those who had been quarantined; however, these resulted in being common reactions to discrimination during the outbreak due to ethnicity, country of origin, and health status prior to the pandemic [6].

A later literature review [8] and studies about the impact of the current COVID-19 outbreak and related restrictions [3, 9–12] found that psychological responses were similar to those to previous outbreaks: Anxiety and depression were the main indicated and detected ones, followed by stress, distress, insomnia, and post-traumatic stress symptoms [8]. These studies found that being women, nurses, of a younger age, or at high risk of contracting COVID-19, as well as having lower socioeconomic status, social isolation, and spending more time watching or reading news about the pandemic, were major risk factors for adverse outcome [7, 8, 10–12]. Other factors reported as causing risk were being single and having a high sexual drive and riskier sexual behavior before the pandemic [13].

Maintaining contact with friends and family and receiving social support were found to be essential in reducing feelings of loneliness and frustration [4, 6]. Other effective coping responses acted upon before and during the pandemic were taking control of one's health status by applying behaviors to protect oneself and others, taking a positive attitude toward the overall situation, and seeking distractions in daily activities [6, 8, 10–12]. For instance, the use

of pornography, phone sex, and webcam sex would help people to reduce stress due to the state of uncertainty and insecurity [13].

The lockdown period may certainly have induced important changes in social relations because of the paradox of the imbalance of spatial distances related to interpersonal communicative interaction (proxemics), distorted both by physical distance—since social contact was possible only to the extent that a certain physical/social distancing was guaranteed—and by too much proximity, i.e. living together forced by the restriction of movement. Of course, social distancing slows down the spread of the virus, but it also leads people to reformulate their social and sexual life [2].

Last April (2021), the famous international journal *The Archives of Sexual Behavior* launched a call for a special issue on COVID-19 in which space could be given to interventions on the impact of the pandemic on sexual health and behavior, sexuality and sexual relationships, access to healthcare and treatment, and the sexual and reproductive rights of all individuals. The same call was also made by other journals such as *Sexuality and Disability* and *Sexually Transmitted Infections* on topics such as prevention, sexuality and disability, access sexual healthcare centers, and sexual behavior [14, 15].

A review conducted by Döring [16] on recent global media narratives (print, radio, and television, Facebook, YouTube, Instagram, and Twitter) and scientific observations and predictions about sexuality-related effects of COVID-19 highlighted at least two aspects of sexual behavior and sexual and reproductive health that may have been affected by the pandemic. And not necessarily negative ones. Döring [16] stated that it is gratifying that issues of sexual and reproductive health and rights have entered mainstream media content so fast. It is noteworthy that technology-mediated sexual intercourse and masturbation with pornography and sex toys have been so easily normalized, to the point where they are officially recommended by the media and health authorities as preventive health behaviors [16, NYCHD, 17]. These behaviors are likely to involve young adults more because of their greater familiarity with the use of electronic devices and social media [18–20].

Other positive effects were found by a group of Southeast Asian researchers [21] on the frequency of sexual intercourse and the improvement in emotional bonding in married couples during lockdown compared to the period immediately before. It was also found that the happiness of married individuals could be slightly increased by isolation [2]. For similar reasons, given the increased "marital fidelity" and the difficulty of having casual sex, reductions in risky sexual behaviors could affect the incidence of sexually transmitted infections [22]. In addition, even if some people violated social-distancing restrictions to see their partners, they tried to minimize physical contact with them in order to reduce possible exposure to the virus [13].

Not all researchers seem to observe only positive effects on sexual health. In fact, the Chinese research group led by Li and colleagues [22] found a reduction in sexual satisfaction and desire during the quarantine period in almost a third of the young Chinese people surveyed. Ferrucci et al. [3] showed similar results for the Italian general population: Decreased sexual activity emerged as one of the daily activities most affected by the psychological impact of COVID-19. However, Panzeri et al. [23], Schiavi et al. [7], and Yuksel and Ozgor [24] found that most of the couples who participated in their studies reported few differences in their sexual life compared to the period prior to the outbreak of the pandemic. In all these studies, female participants were the ones who reported major changes. These were described as a decrease in the quality of pleasure, satisfaction, desire, and arousal/excitation; the reasons they gave were related to the psychological impact of the pandemic and restrictions, such as worry, lack of privacy, and stress [23]. Negative effects of stress are evident in female quality of life, especially regarding sexual functions. It is proven that during the COVID-19 epidemic, Italian women living with their partners suffered from negative influences from the external

environment and that the emergency status impacted emotionally on women's psychology [7]. An effect of this emotional condition was found in a sample of Turkish women by Yuksel and Ozgor [24]. Despite an increase in the frequency of intercourse and an increase in sexual desire, the number of women participating in the study (N = 58) who intended to become pregnant decreased from 32.7% to 5.1% in the face of a significantly decreased rate of contraception use by women; moreover, menstrual disorders were more common than before [24].

The present study reported quantitative and qualitative data on the sexual health and behaviors (activity, desire, arousal, fantasies, masturbation, use of sexual aids, emotional awareness) of Italian adults collected at the end of the lockdown period in May 2020. To the best of our knowledge, no other research has been conducted on sexual health and life during the pandemic using multimethod research, by comparing data from self-administered closed-ended questionnaires and short text messages (250 characters) in response to open-ended questions.

We next present our expectations about the association between the lockdown condition and sexual health and behaviors of Italian adults.

## Expectations

Our expectations assume that the lockdown situation experienced in the first months of the pandemic may have altered various aspects of Italians' sexuality, due to the changes in daily life imposed by the current COVID-19 outbreak.

Since people could have more time to reflect on the role of sexuality in their own experiences—due to restrictions on social participation and limitations in work activities, as well as the increased time they had to spend in their homes, isolated or in forced cohabitation—we predicted that participants would show a growing consciousness about meanings, implications, and expectations regarding sexuality. In addition, negative feelings affected by anxiety and depression, due to the terrible outbreak of the pandemic that particularly burdened Italy before any other Western country, could affect sexual interest and arousal.

Our expectations match with both Arafat et al.'s study [21]—which reported an increase in the occurrence of sexual intercourse and an improvement in affective attachment in couples married during the lockdown compared to the period immediately before—and that of Panzeri et al. [23], which reported a negative impact on sexual habits because of the lack of privacy and the constant closeness. Moreover, we expected that the constant and demanding presence of children at home would have reduced sexual desire and arousal [7]. At the same time, we expected that having to cope with a period of separation or the inability to meet new partners would have prompted participants to resort to the use of sexual aids and to adopt alternative and creative ways to have sex.

As Li, Li [22], Li et al. [22], and Schiavi et al. [7] found in regard to the decrease in sexual satisfaction and desire during the lockdown period in nearly a third of young Chinese and Italian women, respectively, as well as the increase in sexual arousal and desire as reported by Cocci et al. [25], we predicted that we would also find a nonunique sexual behavioral response among our participants. We felt that personal beliefs and feelings about the pandemic and sexuality might have different impacts on sexual arousal and desire.

In summary, we hypothesize positive relationships between the following factors:

1. Negative feelings related to anxiety and depression might affect sexual interest and satisfaction, leading to a more conscious way of experiencing sexual life and habits;

2. Restrictions on participation and limitations on activities might affect the frequency and perceived quality of sexual relations;

3. Changes in sexual intercourse and masturbatory activities may have occurred with the implementation of sexual tools and toys;

4. The lockdown period could lead to modifications in sexual desire and arousal.

## Materials & methods

### Study design

We adopted a cross-sectional survey design conducted with a mixed qualitative-quantitative method. In the sample, variables within dependent type were analyzed. The rationale of this study was descriptive. Data were collected through a mixed-method study with open- and closed-ended questions.

### Materials and apparatus

**Open-ended e-questionnaire on sexual experience.** Participants were invited to respond with a maximum of 280 characters ("tweets") to five open-ended questions formulated in Italian on the following five topics:

1. sexual activity ("Have you experienced a change, for the better or worse, in your sexual activity during the lockdown period? Define it with three adjectives, and, for each one, add or specify something more.");

2. desire, arousal, and fantasies ("How did you do with your sexual desire and arousal? Have you noticed a change in your erotic fantasies?");

3. masturbation ("How did you experience your masturbation during the lockdown period? How did the mode and amount of autoeroticism change, if at all, during quarantine?");

4. use of sexual aids ("What experience have you had during this lockdown period with sex toys or other sexual aids, such as sexually explicit materials and websites? Have you used technological aids to engage in online sex individually or with your partner (e.g., sexting)?");

5. awareness about sexuality ("With respect to your sexuality and your experience of it in the period just past, what do you feel you have understood that is important to you? Have you discovered new sensations with respect to your body and your sexuality? What would you keep and what would you leave behind?").

**Sexual Modes Questionnaire (SMQ)–Nonbinary form.** This is a closed-ended self-report questionnaire assessing the interactions between cognitions, emotions, and sexual response. In its original English version [26], the questionnaire is composed of three independent subscales, available in a female and male version: (i) "Automatic Thought" (AT), (ii) "Emotional Response" (ER), and (iii) "Sexual Response" (SR). The three subscales are composed of 30 items (male) or 33 items (female). For each item, the respondents first have to indicate for the AT subscale through a five-point Likert-type scale the frequency with which a thought or an image occurs (from 1 = "never" to 5 = "always"). Then, for the same item answered in the AT subscale, which has been assigned a value $\geq 2$ (i.e., other than "never"), the respondents assess emotions that they experience during sexual activity in the ER subscale. Respondents are given a list of 10 emotions, of which eight are negative (worry, sadness, disillusion, fear, guilt, shame, anger, hurt) and two positive (pleasure and satisfaction), to select from in evaluating their

responses to the AT items. Finally, in the SR subscale, respondents assess subjective sexual responses pertaining to the items of the AT subscale (which has been assigned a value of $\geq 2$) through an additional five-point Likert-type scale (from 1 = "very low" to 5 = "very high"). The validity and reliability of the Italian version of the test have recently been demonstrated by a psychometric study conducted by Nimbi, Tripodi [27]. In our questionnaire, a modified version of the 56-item unique nonbinary (male/female) form of the SMQ was used, obtained from merging the Italian version of the 30-item male and 33-item female forms after eliminating duplicate items. The Likert-type scales were anchored as the original: the higher the score, the higher the frequency of AT and arousal of SR.

**Sociodemographic and behavioral e-Questionnaire during quarantine.** Two types of information about participants were collected anonymously: sociodemographic data and information about daily living during the first lockdown period. The sociodemographic section inquired about: (i) sex as assigned at birth, (ii) age, (iii) gender identity ("I see myself/ define myself as a man"; "I see myself/define myself as a woman", plus the 56 gender options drawn from Facebook for respondents who did not simply identify as a "man" or "woman"), (iv) sexual orientation based on the Kinsey Scale [28, 29], (v) partner, (vi) marital status, (vii) education level, (viii) employment, (ix) political orientation, (x) religious affiliation, (xi) disability/nondisability of the respondent, and (xii) disability/nondisability of child(ren). The section on daily life experience during the lockdown period surveyed explored: (i) with whom the lockdown period was experienced, (ii) activity and work pattern during the lockdown period, (iii) province in which one lived during the lockdown, (iv) maintenance of a romantic relationship during the lockdown, (v) change in sex life during the lockdown, and (vi) exposure to the COVID-19 virus.

The survey could only be accessed through the Qualtrics.xm (Provo, UT, USA) Internet platform. Each time a questionnaire was completed, the data were sent directly to the online database. Microsoft Excel software was then used for entering the data obtained and IBM-SPSS version 25 software for their statistical processing. Qualitative processing and coding were assisted by ATLAS.ti version 8.4.24 software.

## Procedures

The administration of the questionnaire "Sexuality in Quarantine" took place from May 5 to 14, 2020—i.e., immediately after the lockdown ended and business reopened [30]. Adults ($\geq 18$ years old) residing in Italy during the lockdown, understanding the Italian language, and owning a device with Internet access, by clicking on a link available on the main social media (Facebook, Instagram, Twitter), could gain access to the digital questionnaire called "Sexuality in quarantine." After accessing the e-questionnaire, the software showed the following pages: (i) information sheet and brief introduction to the study, (ii) informed consent, and (iii) privacy policy and processing of personal data. The participants could continue filling out the e-questionnaire by indicating that they had read the consent information. Respondents could decide to withdraw at any time without any penalty.

The survey was structured with the three questionnaires in the following succession: open-ended questionnaire; Sexual Modes Questionnaire (SMQ) in nonbinary form; sociodemographic and behavioral questions that were required to be completed, otherwise the test would be canceled.

Responses to the questionnaires were collected anonymously: participants were assigned an anonymous alphanumeric protocol code, automatically generated by the Qualtrics.xm platform, and used by researchers for data processing. The questionnaire could be completed in an estimated time of 20 minutes.

## Statistical analyses

Data were analyzed using a mixed methodology including both a quantitative and a qualitative approach. All research data are available in a public repository [31].

**Quantitative analyses.** These were conducted on participants' responses covering the SMQ questionnaire. Descriptive analyses (mean, mode, median, standard deviation, frequencies, minima and maxima) and parametric inferential analysis (*t*-test, chi-square, Cronbach's alpha, and one-way ANOVA) were performed.

**Qualitative analyses.** ATLAS.ti 8 software was used to process all respondents' open-ended tweets. Using this qualitative text analysis tool, it was possible to extract only those words (N = 1,313) that were relevant to the questions asked and had semantic relevance to the study. Specifically, entering all tweets (2,315) into ATLAS.ti yielded a set of 4,535 words. From this raw list, we proceeded to extract a "stoplist," i.e. a set of empty words (articles, pronouns, conjunctions, etc.). Then, to facilitate the analysis, the stemming procedure (manually conducted and not supported by software) was used to reduce inflected (or derived) words to their word stem (i.e., the main part of a word that stays the same when endings are added to it). Whole words (i.e., not the stem) were maintained when present only once in the list. The 1,313 words not belonging to the stoplist were associated with each other based on the stem of each word, obtaining final list of 607 stems (S1 Table). The Term Frequency-Inverse Document Frequency [TF-IDF; 32] weight function was applied to the list of stems. TF-IDF allowed us to measure the relevance that a word takes in its context of use. There are five contexts of use in this study, each of which collects all tweets in response to each of the five questions (sexual activity; desire, arousal, and fantasies; masturbation; use of sexual aids; awareness of sexuality). This measure of relevance is from the computation of inverse correlation between the frequency of a word stem among the tweets provided in response to each question and the frequency of the same stem among all 2,315 tweets gathered. Given a collection $C$ of documents $d$, the TF-IDF value for each term $t$ in a document $d \in C$ is calculated as:

$$\mathrm{TF\!-\!IDF}(t, d, C) = \mathrm{TF}(d, t) * \mathrm{IDF}(t, C),$$

where TF $(d, t)$ indicates the number of times a target term $t$ appears in document $d$, and IDF is equal to *log(N/n)*, where *N* indicates the number of documents in $C$ and *n* the number of documents where $t$ is used. In this study case, the TF part of the formula, the collection $C$ is the total number of documents ($C$ = 5), the term $t$ corresponds to each of the roots (e.g., "abbracci*"), and the document $d$ is equal to each one of the five questions belonging to the collection $C$. To avoid favoring longer documents, the TF has been divided by the length of the document itself, where the latter is considered as the total number of stems for each open-ended question. Concerning the IDF formula instead, *N* is equal to the number of open-ended questions ($N$ = 5) and *n* is equal to the number of questions where the stem appears. The salience of a stem (relative weight or TF-IDF) was considered higher the more its frequency in a specific context of use was inversely proportional to its frequency within the total number of 607 stems (i.e., the final list computed; S1 Table).

Stems above the first (positive) standard deviation of the TF-IDF scores of each group of stems were selected for each of the five open-ended questions. (As none of the stems belonging to the fourth open-ended question exceeded the cut-off [SD $\geq$ 1], these were excluded from further analyses). Subsequently, the stems were hierarchically clustered based on Euclidean distances with respect to TF-IDF scores for each remaining context of use. Subsequently, the stems were hierarchically clustered based on Euclidean distances with respect to TF-IDF scores for each context of use (S2–S5 Tables). For the choice of cluster solution (i.e., number of final clusters), the variation in the agglomeration coefficient was observed (S6–S9 Tables). This

represents the degree of inhomogeneity within the cluster each time a new item (or cluster of items) was merged: The higher the coefficient, the more dissimilar the grouped items were, and the more the inhomogeneity increased. Therefore, the solution preceding the maximum variation of the agglomeration coefficient was chosen as the best explanation for the final number of clusters.

### Ethical issues

The project was approved by the Committee of Bioethics of the University of Perugia, protocol no. 51854/2020. The observational study was carried out with full respect for the dignity of the human being and his/her fundamental rights, as dictated by the Declaration of Helsinki and the rules of Good Clinical Practice issued by the European Council. Informed consent was obtained from all individual participants included in the study. Written informed consent was obtained from all participants 18 years and older.

## Results

### Participants

A total of 465 people responded to the survey (sex as assigned at birth: female = 78.7%). The mean age was 29 ± 10.37 years (range = 18–34). Most participants reported having at least a bachelor's degree (71.2%). In regard to political orientation, the majority stated that they were "left" (27.1%). "Center" affiliation was stated by 9.5%, "right" by 7.1%, while 41.7% reported "no" affiliation or "I don't know." The top two, and nearly exclusive, groups in which participants placed themselves with respect to religious affiliation were Catholicism (44.3%) and no religion (46%). Some 97.2% of the participants identified themselves as having a binary male/female gender identity. As regards sexual orientation (Kinsey Scale), 71.6% of the respondents affirmed they were exclusively heterosexual, 2.4% exclusively homosexual, and 3.4% bisexual. In terms of functioning and disability, 5.6% of the respondents reported that they had a disability, with the majority (5.1%) having an invisible or sensory disability. Most respondents (41.9%) lived in the municipality of Perugia during the quarantine period. A total of 396 participants (85.1%) reported that they had not been exposed to COVID-19. All data collected by the sociodemographic and behavioral e-questionnaire are reported in S10 Table.

### Quantitative analysis on SMQ

All responses to the first and third scales of the SMQ were split quintiles. To discriminate between major and minor values assigned to each item, only items belonging to the first quintile (items with lower scores) and fifth quintile (items with higher scores) were extracted. Respondents to the first scale (AT) had access to the second (ER) and third scales (SR) only if the item on the first scale was assigned a score $\geq$ 2 (i.e., other than "never"). Cronbach's alpha was computed to test the reliability of the two Likert-type scales, obtaining excellent internal consistency: AT scale on 56 items, $\alpha$ = .915; SR scale on 56 items, $\alpha$ = .917.

t-tests and one-way ANOVAs were conducted to compare the effect of the variables collected through the sociodemographic questionnaire (independent variable) on the response values that participants attributed to the SMQ questionnaire scales (dependent variable). Effects were found with regard to the t-test from sex as assigned at birth (male/female) and with the ANOVA from disability/nondisability on the SMQ questionnaire. Only the results of these two analyses are reported below. Where appropriate, Bonferroni's corrections have been applied.

**Automatic Thought (AT), Emotional Response (ER), and Sexual Response (SR).** The first scale of the SMQ concerns AT. These are the thoughts about one's beliefs, judgments, and moods, both positive and negative, that arise when one is exposed to sexual stimuli. These thoughts are often good predictors of sexual desires. Out of the total number of participants (N = 465), the automatic thoughts that received the highest mean value on the response scale, where 1 indicated "never" and 5 indicated "always" (max = 4.23, min = 3.35, i.e. between "sometimes" and "always"), were Items 5, 6, 10, 12, 20, 21, 23, 27, 31, 38, 39, 41, 46, and 50.

Items 12, 39, and 50 result in a higher mean value in the responses of females only, and not in males. In males, but not in females, two items had higher mean values, Item 6, "I am the happiest person in the world" (mean = 2.86), and Item 31, "I need to show my manhood/femininity" (mean = 2.61). Also, compared to the total sample, the 10 automatic thoughts that received the lowest mean scores on the response scale (min = 1.24, max = 1.58, i.e. between "never" and "rarely") were Items 42, 9, 3, 36, 53, 17, 19, 40, 48, and 54.

Among the 10 emotions of the second scale (ER)—of which eight were negative (worry, sadness, disillusion, fear, guilt, shame, anger, hurt) and two positive (pleasure and satisfaction) —associate at each item of the AT scale, positive emotions were found to be more frequent than negative emotions. Out of the eight negative emotions, "worry" (9.30%) emerged significantly as the most frequent and "hurt" (1.96%) as the least frequent, both in males and females. Out of the two positive emotions, "pleasure" (11.50%) was the most frequently selected, turning out to be the most frequently chosen among all other emotions selected, both in males and females.

In the last scale (SR), the mean intensity of sexual arousal that each thought elicited in respondents was 1.42 (DS = .56), i.e. between "very low" and "low." The items belonging to the fifth quintile were 5, 12, 20, 21, 23, 27, 31, 39, 41, and 50, with Item 12 ("Making love is wonderful") obtaining the highest mean score (4.04). Males reported a higher level of arousal (1.47) than females (1.40). The items belonging to the first quintile were 3, 9, 17, 19, 25, 32, 33, 34, 35, 36, 37, 40, 42, 45, 48, 52, 53, 54, 55, and 56, with Item 45 ("I am not penetrating my partner") obtaining the lowest mean score (.40).

**Sex differences on SMQ scales.** A t-test was computed to investigate the effect of the independent variable sex (male or female based on the answers to "sex as assigned at birth") on the occurrence of automatic thoughts (AT scale, dependent variable), indicated by respondents through a five-point Likert-type scale anchored by 1 (never) to 5 (always). As displayed in Table 1, significant differences in values were found among 13 items of the first scale AT.

**Disability effect on SMQ scales.** One-way ANOVAs were conducted to compare the effect of the variables collected via the sociodemographic questionnaire (independent variable) on the occurrence of automatic thoughts (AT scale, dependent variable), indicated by respondents through a five-point Likert-type scale anchored by 1 (never) to 5 (always). Only the variable "xi. disability/nondisability" generated a statistically significant difference, $F(1, 5.944, p < .05)$. A univariate ANOVA with marginal mean estimation was computed on the 56-item AT scale. As shown in Table 2, significant differences in values among 17 items were found.

The same ANOVA statical method was also computed on the SR scale. As shown in Table 3, significant differences in values among seven items were found.

A chi-square test was computed on the ER scale. As displayed in Table 4, significant differences in values among 12 items were found.

## Qualitative analysis of the five open questions

Before proceeding with the qualitative analysis, the authors (ACM and EZ) evaluated the pertinence of responses to the five questions. In agreement with all authors, the evaluators adopted

**Table 1. t-test computed to compare the effect of sex (male/female) on the values of the AT scale at $p < .05$ level.**

| SMQ–Automatic Thought Item | Sex | Mean Differences | Std. Error | Sig. | 95% Confidence Interval for Difference | |
|---|---|---|---|---|---|---|
| | | | | | Lower Bound | Upper Bound |
| **1.** I can't hear anything | Male | 1.9192 | 1.46138 | .036 | -.60460 | -.02166 |
| | Female | 2.2323 | | | | |
| **5.** My body excites him/her | Male | 3.3434 | 1.53808 | .010 | -.71081 | -.09728 |
| | Female | 3.7475 | | | | |
| **9.** He/she is abusing me | Male | 1.4545 | 1.00278 | .038 | .01212 | .41212 |
| | Female | 1.2424 | | | | |
| **12.** Making love is wonderful | Male | 4.3838 | 1.31950 | .009 | .09037 | .61671 |
| | Female | 4.0303 | | | | |
| **14.** If I don't reach orgasm now, I won't be able to later | Male | 1.5657 | 1.43684 | .001 | -.85223 | -.27908 |
| | Female | 2.1313 | | | | |
| **28.** When does it end? | Male | 1.8081 | 1.55070 | .0041 | -.63251 | -.01395 |
| | Female | 2.1313 | | | | |
| **32.** If I let myself go he/she will think I am easy | Male | 1.4141 | 1.43081 | .016 | -.63890 | -.06817 |
| | Female | 1.7677 | | | | |
| **33.** I have to be able to have a relationship | Male | 2.2525 | 1.86351 | .022 | .06267 | .80601 |
| | Female | 1.8182 | | | | |
| **38.** I don't want to be hurt emotionally | Male | 2.4545 | 2.44486 | .001 | -1.37651 | -.40127 |
| | Female | 3.3434 | | | | |
| **41.** What must he/she be thinking about me? | Male | 2.6970 | 1.93829 | .032 | -.81083 | -.03766 |
| | Female | 3.1212 | | | | |
| **43.** I should wait for him to make the first move | Male | 2.0101 | 1.90016 | .002 | -.97494 | -.21698 |
| | Female | 2.6061 | | | | |
| **44.** I am not getting excited | Male | 1.7475 | 1.34288 | .001 | -.75268 | -.21701 |
| | Female | 2.2323 | | | | |
| **46.** I am getting fat/ugly | Male | 1.9798 | 1.82082 | .001 | -.99952 | 9.34670 |
| | Female | 2.6162 | | | | |

a broad criterion of inclusivity to consider an answer pertinent to the question. Only those tweets that were clearly intended to criticize the study or its authors, to insult or threaten people, or that are meaningless (e.g., where only meaningless filler characters have been included in the response field such as "wbkjwiudjeoq") would have been excluded. A total of 2,311 out of 2,325 expected tweets were evaluated as being pertinent to the question posed and uploaded in ATLAS.ti. As detailed above in the subsection "Statistical Analyses," out of 4,535 words collected from tweets, a final list of 607 stems was extracted S1 Table. After applying the TF-IDF [32] weight function, the most salient stems were considered those above the first (positive) standard deviation to the relative TF-IDF mean value. From the final list of 607 (S1 Table), 76 salient stems were extracted as displayed in Table 5.

**First question: An overview of sexual activity during the lockdown.** For the first question ("Did you experience a change, for the better or worse, in your sexual activity during the lockdown period? Define it with three adjectives and for each one, add or specify something more"), 31 salient stems (Table 5) were clustered in three groups (Fig 1).

Based on the common meaning emerging from the stems belonging to the group, the three clusters were named as follows: Cluster 1 "suffering," Cluster 2 "emotional-sexual flattening," and Cluster 3 "feelings about sexual activity."

**Table 2. Significant univariate effects for presence and absence of disability on the values of AT scale at p < .05 level.**

| SMQ–Automatic Thought Item | Presence/Absence of Disability | Mean Differences | Std. Error | Sig. | 95% Confidence Interval for Difference | |
|---|---|---|---|---|---|---|
| | | | | | Lower Bound | Upper Bound |
| **3.** It would be better to die than to be like this | Disability | 1.769 | .178 | .031 | .034 | .734 |
| | Nondisability | 1.385 | | | | |
| **9.** He/she is abusing me | Disability | 1.769 | .14 | .002 | .167 | .715 |
| | Nondisability | 1.328 | | | | |
| **11**. He/she doesn't find my body attractive anymore | Disability | 2.307 | .211 | .047 | .005 | .834 |
| | Nondisability | 1.888 | | | | |
| **13.** I'm condemned to failure | Disability | 2.269 | .213 | .022 | .069 | .907 |
| | Nondisability | 1.781 | | | | |
| **16.** I am only doing this because he/she asked me to | Disability | 2.230 | .197 | .026 | .054 | .827 |
| | Nondisability | 1.790 | | | | |
| **17.** He/she is not respecting me | Disability | 2.153 | .165 | .001 | .386 | 1.033 |
| | Nondisability | 1.443 | | | | |
| **19.** He only loves me if I'm good in bed | Disability | 2.115 | .189 | .001 | .282 | 1.024 |
| | Nondisability | 1.545 | | | | |
| **25.** He/she only wants to satisfy himself | Disability | 2.307 | .212 | .007 | .153 | .986 |
| | Nondisability | 1.464 | | | | |
| **33.** I must be able to have intercourse | Disability | 2.692 | .261 | .001 | .33 | 1.355 |
| | Nondisability | 1.849 | | | | |
| **35.** How can I get out of this situation? | Disability | 2.384 | .255 | .036 | .036 | 1.038 |
| | Nondisability | 1.847 | | | | |
| **36.** This is disgusting | Disability | 1.769 | .151 | .016 | .07 | .662 |
| | Nondisability | 1.403 | | | | |
| **37.** Sex is all he/she thinks about | Disability | 2.269 | .219 | .013 | .115 | .975 |
| | Nondisability | 1.724 | | | | |
| **38.** I don't want to get hurt emotionally | Disability | 3.576 | .323 | .033 | .056 | 1.326 |
| | Nondisability | 2.885 | | | | |
| **48.** If I refuse to have sex, he/she will cheat on me | Disability | 2.153 | .194 | .001 | .259 | 1.023 |
| | Nondisability | 1.512 | | | | |
| **51.** He/she will replace me with another guy | Disability | 2.153 | .238 | .013 | .128 | 1.062 |
| | Nondisability | 1.981 | | | | |
| **52.** I'm getting old | Disability | 2.500 | .228 | .001 | .282 | 1.178 |
| | Nondisability | 1.770 | | | | |
| **55.** This is not going anywhere | Disability | 2.384 | .246 | .033 | .042 | 1.01 |
| | Nondisability | 1.858 | | | | |

The first cluster, "suffering," consisting of only one item, highlights a general experience of suffering due to the limitations that the lockdown brought with it. For instance, a cisgender woman aged 25 wrote about her stress condition:

"Physically and mentally I suffer from a lack of intimacy and bodily interaction."

Items in the second cluster refer to a climate of "emotional-sexual flattening," evidencing how the restriction to mobility experienced by participants and the uncertainty of the future negatively affected sexual experiences and mood. For instance, a 24-year-old man wrote about his depression:

**Table 3. Significant univariate effects for presence and absence of disability on the values of the SR scale at p < .05 level.**

| SMQ–Automatic Thought Item | Presence/Absence of Disability | Mean Differences | Std. Error | Sig. | 95% Confidence Interval for Difference | |
|---|---|---|---|---|---|---|
| | | | | | Lower Bound | Upper Bound |
| **3.** It would be better to die than to be like this | Disability | .884 | .199 | .014 | .101 | .884 |
| | Nondisability | .391 | | | | |
| **9.** He/she is abusing me | Disability | 1.153 | .228 | .004 | .210 | 1.105 |
| | Nondisability | .496 | | | | |
| **12.** Making love is wonderful | Disability | 3.500 | .227 | .013 | -1.010 | -.118 |
| | Nondisability | 2.936 | | | | |
| **17.** He/she is not respecting me | Disability | 1.307 | .218 | .001 | .293 | 1.151 |
| | Nondisability | .585 | | | | |
| **19.** He only loves me if I'm good in bed | Disability | 1.230 | .239 | .007 | .176 | 1.115 |
| | Nondisability | .585 | | | | |
| **33.** I must be able to have intercourse | Disability | 1.615 | .273 | .008 | .190 | 1.264 |
| | Nondisability | .888 | | | | |
| **45.** I am not penetrating my partner | Disability | .846 | .207 | .024 | .061 | .875 |
| | Nondisability | .378 | | | | |

"Often practicing autoerotism, I stopped, blocked, and dwelt on thinking about anything else, as if it was an obligation."

The third cluster, "feelings about sexual activity," includes terms about how participants felt and acted out sexuality under restricted social conditions.

"[I experienced] a change for the worse that can be described as: frustrating in that the desire remained unexpressed and unfulfilled." (Gender fluid, 26 years old)

"[I would define my sexual experience as] comfortable in the sense that I had the ability to choose when and how to do it with no time or space limitations." (Woman, 29 years old)

"The only times I masturbated I was anxious because I was living with my parents and I didn't feel comfortable." (Woman, 31 years old)

**Table 4. List of items showing a significant association between disability condition (presence/absence) and types of emotion (negative and positive) scale at p < .05 level.**

| SMQ–Automatic Thought Item | χ(df) | Sig. |
|---|---|---|
| **9.** He/she is abusing me | 13.581 | .001 |
| **16.** I am only doing this because he/she asked me to | 11.509 | .003 |
| **17.** He/she is not respecting me | 13.097 | .001 |
| **19.** He only loves me if I'm good in bed | 12.427 | .002 |
| **25.** He/she only wants to satisfy himself | 12.830 | .002 |
| **33.** I must be able to have intercourse | 8.162 | .017 |
| **35.** How can I get out of this situation? | 16.585 | .000 |
| **38.** I don't want to get hurt emotionally | 15.894 | .000 |
| **4.** He/she only cares about me when he/she wants to have sex | 7.480 | .024 |
| **45.** I'm not penetrating my partner | 9.712 | .008 |
| **48.** If I refuse to have sex, he/she will cheat on me | 7.229 | .027 |
| **49.** If he/she just whispers something romantic in my ear | 2.614 | .271 |

**Table 5. Frequency data about the salient stems above the first (positive) standard deviation of the TF-IDF scores for each of the five open-ended questions.**

| Stem | F | TF | IDF | TF-IDF | Stem | F | TF | IDF | TF-IDF |
|---|---|---|---|---|---|---|---|---|---|
| | | | | **Open-Ended question 1 (N = 31)** | | | | | |
| Suffer* | 6 | .00248 | 1.60944 | .00399 | Calm* | 8 | .00330 | .51083 | .00169 |
| Depress* | 8 | .00330 | .91629 | .00303 | Relax* | 18 | .00743 | .22314 | .00166 |
| Asexual* | 4 | .00165 | 1.60944 | .00266 | Passion* | 17 | .00702 | .22314 | .00157 |
| Repress* | 4 | .00165 | 1.60944 | .00266 | See* | 4 | .00165 | .91629 | .00151 |
| Listless* | 4 | .00165 | 1.60944 | .00266 | Affect* | 7 | .00289 | .51083 | .00148 |
| Intens* | 26 | .01074 | .22314 | .00240 | Abstinence | 7 | .00289 | .51083 | .00148 |
| Nervous* | 6 | .00248 | .91629 | .00227 | Satisf* | 15 | .00620 | .22314 | .00138 |
| Anxi* | 22 | .00909 | .22314 | .00203 | Surrender* | 2 | .00083 | 1.60944 | .00133 |
| Comfort* | 3 | .00124 | 1.60944 | .00199 | Contrasting | 2 | .00083 | 1.60944 | .00133 |
| Confused | 3 | .00124 | 1.60944 | .00199 | Impotence | 2 | .00083 | 1.60944 | .00133 |
| Innovativ* | 3 | .00124 | 1.60944 | .00199 | Unwanted | 2 | .00083 | 1.60944 | .00133 |
| Loneliness | 9 | .00372 | .51083 | .00190 | Insatiable | 2 | .00083 | 1.60944 | .00133 |
| Happy* | 5 | .00207 | .91629 | .00189 | Argu* | 2 | .00083 | 1.60944 | .00133 |
| Melancholy* | 5 | .00207 | .91629 | .00189 | Reinvent* | 2 | .00083 | 1.60944 | .00133 |
| Frustr* | 20 | .00826 | .22314 | .00184 | Relief | 2 | .00083 | 1.60944 | .00133 |
| Sad* | 20 | .00826 | .22314 | .00184 | | | | | |
| | | | | **Open-Ended Question 2 (N = 7)** | | | | | |
| Dream* | 13 | .00926 | .51083 | .00473 | Transgress* | 3 | .00214 | .91629 | .00196 |
| See* | 4 | .00285 | .91629 | .00261 | Extreme* | 3 | .00214 | .91629 | .00196 |
| BDSM | 4 | .00285 | .91629 | .00261 | Intens* | 9 | .00641 | .22314 | .00143 |
| Variety | 2 | .00142 | 1.60944 | .00229 | | | | | |
| | | | | **Open-Ended Question 3 (N = 10)** | | | | | |
| Privacy | 8 | .00806 | .51083 | .00412 | Relax* | 10 | .01007 | .22314 | .00225 |
| Shower | 3 | .00302 | .91629 | .00277 | Calm* | 4 | .00403 | .51083 | .00206 |
| Loneliness | 5 | .00504 | .51083 | .00257 | Date* | 2 | .00201 | .91629 | .00185 |
| Fatigue* | 5 | .00504 | .51083 | .00257 | Movie* | 2 | .00201 | .91629 | .00185 |
| Intens* | 11 | .01108 | .22314 | .00247 | Leisure | 2 | .00201 | .91629 | .00185 |
| | | | | **Open-Ended Question 4 (N = 12)** | | | | | |
| Vibrator* | 13 | .00727 | .51083 | .00372 | YouPorn | 4 | .00224 | .91629 | .00205 |
| Movie* | 7 | .00392 | .91629 | .00359 | Whatsapp | 2 | .00112 | 1.60944 | .00180 |
| Photo* | 27 | .01511 | .22314 | .00337 | Chat | 2 | .00112 | 1.60944 | .00180 |
| Webcam | 6 | .00336 | .91629 | .00308 | Lubricant | 2 | .00112 | 1.60944 | .00180 |
| Cell phone* | 3 | .00168 | 1.60944 | .00270 | Audio* | 12 | .00672 | .22314 | .00150 |
| Stories | 5 | .00280 | .91629 | .00256 | Pornhub | 5 | .00280 | 0,51083 | .00143 |
| | | | | **Open-Ended Question 5 (N = 16)** | | | | | |
| Underst* | 4 | .00221 | 1.60944 | .00356 | Affect* | 8 | .00443 | .51083 | .00226 |
| Know* | 12 | .00664 | .51083 | .00339 | Kiss* | 4 | .00221 | .91629 | .00203 |
| Emotion* | 6 | .00332 | .91629 | .00304 | Attract* | 7 | .00387 | .51083 | .00198 |
| Foreplay | 6 | .00332 | .91629 | .00304 | Grant* | 2 | .00111 | 1.60944 | .00178 |
| Bonding | 6 | .00332 | .91629 | .00304 | Imperfect* | 2 | .00111 | 1.60944 | .00178 |
| Tenderness | 6 | .00332 | .91629 | .00304 | Patience | 2 | .00111 | 1.60944 | .00178 |
| Orientation | 3 | .00166 | 1.60944 | .00267 | Unlock me | 2 | .00111 | 1.60944 | .00178 |
| Love* | 19 | .01051 | .22314 | .00235 | Dirty | 2 | .00111 | 1.60944 | .00178 |

Row frequencies (F), term frequency (TF), inverse document frequency (IDF), and term frequency-inverse document frequency (TF-IDF) of the most relevant stems extracted from answers to the five open-ended questions are reported. The asterisk indicates that only the stem word has been listed, i.e., a part of a word that can form the basis of other words with similar meaning through the addition of suffixes.

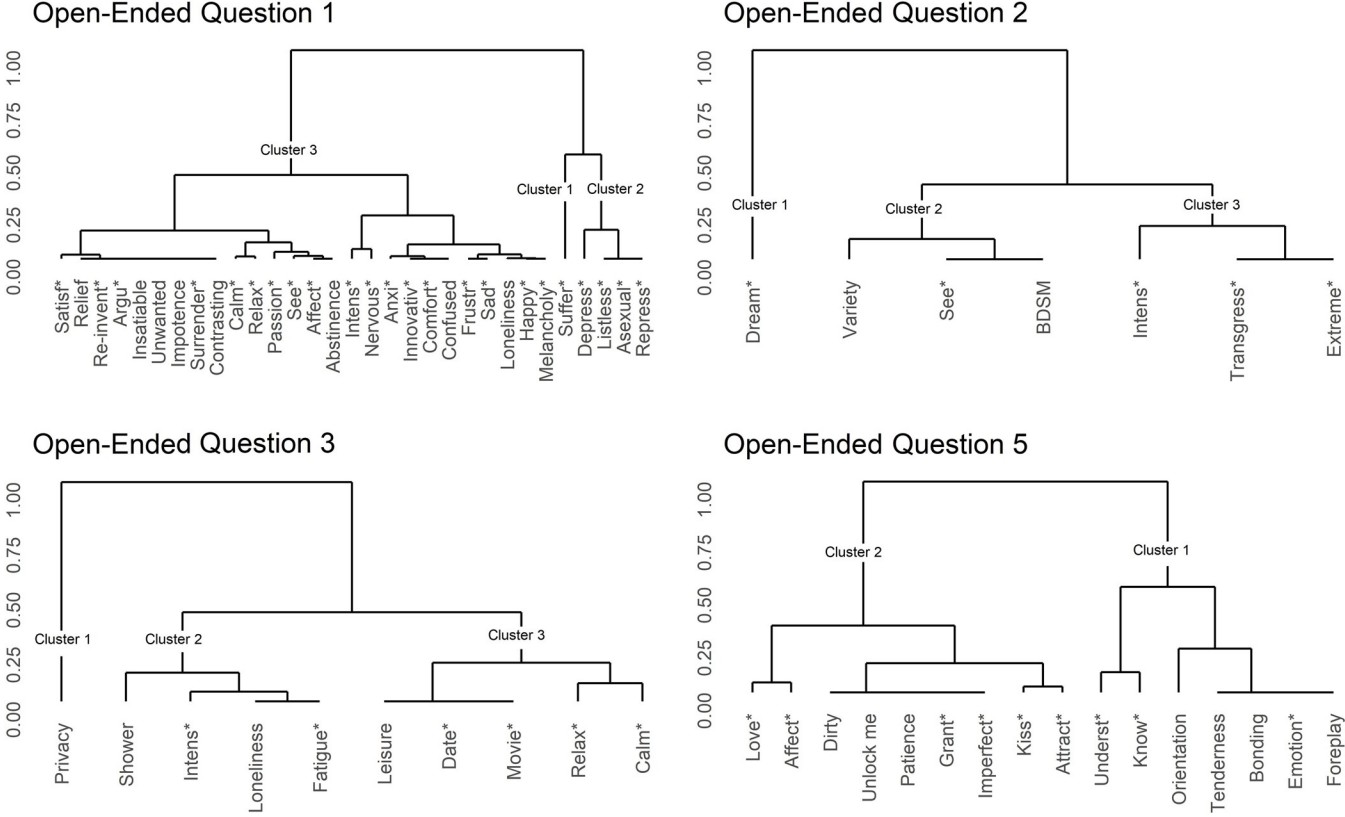

**Fig 1. Cluster dendrograms of extracted terms (open-ended questions).** Cluster dendrograms are based on the Euclidean Distances' Matrix of the 64 Salient Stems for Open-Ended Question 1, 2, 3, and 5. A Hierarchical Cluster Analysis was performed to group similar terms in relation to their TF-IDF Index. Euclidean distances are reported on a modified scale with units from 0 to 1 (0 = maximum proximity/similarity; 1 = maximum distance/dissimilarity).

"I wish to see again as soon as possible the person with whom I like to make love." (Woman, 29 years old)

"Compared to the stress and frustration of the situation, the sexual activity provided some relief." (Woman, 26 years old)

**Second question: Desire, arousal, and fantasies.**   For the question "How did you do with your sexual desire and arousal? Have you noticed a change in your erotic fantasies?" seven salient stems (Table 5) were clustered in three groups (Fig 1). Based on the common meaning emerging from the stems belonging to the group, the three clusters were named as follows: Cluster 1 "dream," Cluster 2 "variety," and Cluster 3 "opportunities for transgression."
The first cluster, "dream", indicates a phenomenon through which fantasies were expressed.
"Both desire and fantasies increased, expressed more during dreams (especially during afternoon rest). Upon awakening, arousal was strong and long-lasting. It happened more rarely before the lockdown." (Woman, 32 years old).
The second cluster, "variety," shows a wide sexual exploration of fantasies by the participants.

"Yes, I started having other fantasies, from threesome to BDSM. I've always thought about it, but during the lockdown I seriously thought about enacting them." (Woman, 24 years old)

The third cluster points out that, for some respondents, fantasies were experienced as an "opportunity for transgression."

"I have discovered and also experienced new fantasies related to the BDSM world and in particular, submission." (Woman, 24 years old)

"[When I was alone] I had moments of great excitement, teenage I would say. I haven't noticed excessive changes, but I have had an acceptance of more 'extreme' fantasies that before I didn't like." (Man, 33 years old)

**Third question: Masturbation.**   For the third question ("How did you experience your masturbation during the lockdown period? How did the mode and amount of autoeroticism change, if at all, during quarantine?"), 10 salient stems (Table 5) clustered into three groups (Fig 1). The three clusters were named as follows: Cluster 1 "privacy," Cluster 2 "need for space," and Cluster 3 "escaping from the room."

The first cluster consists of only one very salient stem, "privacy."

"Unfortunately, my moments of privacy have been abruptly reduced and I haven't had the opportunity to masturbate with the same tranquility and frequency as before." (Woman, 27 years old)

The second cluster refers to the reduction of space and time for solitude and self-eroticism.

"Due to lack of space, privacy, or discomfort, desire has often been experienced with frustration, as an impediment." (Man, 28 years old)

The third cluster indicates the calming and relaxing role of masturbation during the quarantine period, as a way of "escaping from the room."

"The amount of autoeroticism during quarantine has increased due to simply not being able to meet my partner. It has been critical to release some stress, relax, and satisfy my sex drive." (Woman, 21 years old)

**Fourth question: Use of sexual aids.**   None of the stems extracted from the fourth question ["What experience have you had during this lockdown period with sex toys or other sexual aids, such as sexually explicit materials and websites? Have you used technological aids to engage in online sex individually or with your partner (e.g., sexting)?"] exceeded the first standard deviation. For this reason, we did not proceed with cluster analysis. Twelve stems with higher TF-IDF are reported (Table 5), which give information about the variety of aids participants reported using.

"I have purchased and used sex toys for personal masturbation with great pleasure. I was aware of them, but this period led me to equip myself with them for the future as well. I often used technological support such as videos/photos/sexting with more or less close friends." (Man, 25 years old)

"Yes, we purchased a sex toy during quarantine. I didn't own any before. The choice came down to a wearable vibrator that could be managed remotely, so the partner could control it remotely with an app." (Woman, 27 years old)

"Thanks to YouPorn's offering, I initially viewed pornographic content often but later I preferred sexting with my partner." (Woman, 21 years old)

"Pornhub premium free was a major discovery [. . .]." (Man, 45 years old)

**Fifth question: What has been understood about one's sexuality.**    For the last question ("With respect to your sexuality and your experience of it in the period just past, what do you feel you have understood that is important to you? Have you discovered new sensations with respect to your body and your sexuality? What would you keep and what would you leave behind?"), 16 salient stems (Table 5) were clustered in two groups (Fig 1). Based on the common meaning emerging from the stems belonging to the group, the two clusters were named as follows: Cluster 1 "need for intimacy" and Cluster 2 "acceptance of sexuality."

The first cluster, "need for intimacy," expresses the general need to deepen intimacy with one's own sexuality and with one's partner.

"Definitely I understood how necessary a good and beautiful understanding with my partner outside the sexual sphere is, and that it is absolutely necessary to carve out spaces to remember us and give us a common oxygen. And only afterwards to get closer with sex as well." (Woman, 31 years old)

The second cluster, "accepting sexuality," refers to different facets of the same dimension: accepting sexuality both as an aspect of one's person and in its multiple meanings.

"During this period of quarantine with respect to sexuality, I learned to indulge my moods by sharing them with my partner. I felt no shame for the lack of desire, the scarce excitement." (Woman, 29 years old)

"I realized that I had major issues with my sexuality. I've always felt dirty. I think my awareness and change is a sign of growth and wanting to finally be okay. I am accepting that I am a person who can spark sexual interest." (Woman, 25 years old)

## Discussion

### Quantitative results drawn from the sociodemographic questionnaire and the SMQ

The large number of people who participated in the survey in a short period of time, just 10 days, highlights the centrality and relevance of the theme proposed. The social restrictions imposed by the lockdown should have had a relevant effect on affective and sexual dating, especially in those adults and couples who could not live together.

Respondents, the majority of whom were female (generally more willing to engage in introspective inquiries), especially identify with the statement that "making love is wonderful," perceiving their bodies as exciting, as well as the sexual activities enacted. Male participants also shared these feelings, even adding that arousal was even a little more pronounced than before the pandemic. The sexual experience, in a time of quarantine, was experienced with positive characteristics, openness, an unproblematic relationship with the body, and awareness of, and self-reflection on, one's sexuality.

In contrast, the thoughts connected to worry (AT scale) are rather marginal, and even more marginal are those thoughts connected to pain; very few of the respondents recognize themselves as having thoughts with self-evaluative connotations and of disgust and suffering towards the interlocutor and the sexual activities performed.

From the SMQ quantitative data emerge a series of further significant elements in a rather marked way. First, the percentage of respondents who report having a disability (5.6%) mirrors the Italian national average of people with disabilities, which, according to ISTAT (Italian Institute of Statistics) [33], is 5.2%. These data suggest that adults with disabilities pay attention to, and consider relevant, the topic of sexuality. This is by no means taken for granted if we consider that, while adequate operational strategies to educate people with physical or mental disabilities to sexuality and affectivity have been discussed and studied for some time now, their existential condition still suffers from heavy taboos and concrete limitations in this fundamental area of the expression of their humanity [34–37]. Indeed, sexuality represents an essential component of one's human identity, in emotional, physical, and psychological, as well as in ethical and spiritual terms. Therefore, the adherence to the research on people with disabilities and the self-reflection that emerged is particularly relevant. According to our results (Tables 2 and 3), respondents with disability reveal a marked inversion of the trend with respect to the mean of the respondents about the thoughts in which people recognize themselves (AT scale). Respondents with a disability seem to revolve around feeling abused, inadequately respected, viewed more as sexual "objects" than as persons in a relationship, negatively focused on performance and self-image, emotionally vulnerable and inadequately acknowledged in terms of their needs. In face of such their self-image, which is characterized by a certain amount of suffering and low self-esteem, on the other hand, people with disabilities reported their arousal as being higher than the mean.

Thus, we could hypothesize that, among the participants, those who, in relation to their condition of disability, do not consider themselves lovable, even though they perceive arousal, experience as inadequate the movement towards the satisfaction of this need. Experiencing their own image and performance as inadequate, participants with a disability show a particularly sensitivity to the expression of acceptance and approval by a sexual partner, so much so that they are emotionally dependent on them and attribute their own negative image to the partner's behavior. Thus, on the one hand, the painful experience with respect to low self-esteem, linked to disability [36, 38], could be projected onto the partner's behaviors, while on the other hand, the current condition of fragility may make people more vulnerable and sensitive to criticism and rejection.

Human sexuality is often conditioned by the myth of "bodily perfection" [39], which makes sexuality a right believed to be only reserved for people with healthy, perfect bodies but an aberration if desired and experienced by and with people with disabilities [34, 35]. This myth not only informs nondisabled people but influences the thinking and behavior of people with disabilities as well. Their emotional lives are experienced with fear in the face of the "normal" world, where even an "automatic" thought of sexual arousal and desire for contact can be dangerous and disturbing. Imperfection is ugly, not worthy of giving and receiving love, not even thinking about it. When people with a disability have to face the emergence of sexuality, at times resounding and powerful, another stereotype of an opposite sign also comes under stress, which would presume in a person with disability a condition of perpetual infantilization, without eros and without drives [37, 40].

Consequently, the manifestation of sexuality and its normal events (desire, arousal, acceptance, rejection, satisfaction, frustration) can be experienced with greater difficulty and accompanied by painful emotions. Instead of being the moment of the definition of one's identity and freedom, such manifestations of sexuality can become the fracture of this identity, or a painful and heavy identity to manage, perceived by the environment as inadequate, difficult, and undesirable [41]. Much remains to be done on the educational level and for overcoming stereotypes in this field [34, 35]. It is perhaps relevant here to recall that:

"Health promotion is the process of enabling people to increase control over, and to improve, their health. To reach a state of complete physical, mental and social well-being, an individual or group must be able to identify and to realize aspirations, to satisfy needs, and to change or cope with the environment." [WHO, 42]

Sexuality, it is not pleonastic to repeat, is an essential component of health and a right of every human being.

In the sample examined, Catholic affiliation is well below the national average—74.4% according to Ipsos Public Affairs [43]. Addressing a sexually explicit issue seems to attract more people without religious affiliation, because religions have specific teachings about sex that can condemn masturbation or sexual relationships outside of a heterosexual marriage [36, 44, 45].

## Tweeting sexual experience during quarantine: Qualitative results from the five open-ended questions

The qualitative analysis performed on the tweets has revealed the emotional experience of the participants emerged during the lockdown period. It was generally depressive; both in terms of desire and the actual possibility to perform sexual activity as usual.

The expression of desire and arousal seems to have been shifted more to the level of the imagination, where they could be expressed in much broader and freer terms, while also opening to participants spaces of access to transgression and new sexual practices mediated by electronic devices and the Internet. As several surveys have shown [46, 47], some of the most common sexual fantasies, such as multipartner sex, BDSM, novelty, adventure and variety, homoeroticism, and gender-bending, are found at the top even among our participants. Adaptive behaviors of coping with the frustration generated by the lockdown emerge among our participants. In the presence of heavy experiences, participants allowed themselves to accept as a possibility of transformation and change the restrictions made necessary by the pandemic. Thus, a certain flexibility and fluidity in how respondents think about their gender and sexual identity emerges, especially among female participants [47, 48]. The possibility of autoerotism becomes explicit. The open-ended question on this topic was welcomed with openness and a wealth of qualitative details about the experience, while the opportunity for privacy time and space was significantly emphasized, positively for those who experienced the lockdown in solitude and as an active search for concrete possibilities for those who experienced it in cohabitation with others [49]. A similar movement of creativity and exploratory aptitude can be detected by the possibility of enriching autoerotic practices or sexuality with a partner with aids such as sex toys and photo/video images, especially for sexual practices carried out at a distance, a new experience for many people, both within the affective copy and with casual partners. On the whole, what emerges from the answers to the open-ended questions is—in the face of the frustration and depressive climate generated by the lockdown to the detriment of desire and expression of one's habitual sexual life—an opening to new possibilities of expression of sexuality, accompanied by the rediscovery of the value of tenderness and affectivity and a clearer awareness of one's perception of oneself, one's experience of sexuality, one's needs, desires, and margins for active and creative exploration.

In the first question about sexual activity, the stems contained in the three clusters refer predominantly to emotional states rather than to sexual behaviors, suggesting that participants, when asked to provide an overview on their sexual activity during the lockdown period, highlighted that the changes perceived concerned not so much the sexual behavior itself (e.g., frequency of masturbation, intercourse, fantasies, etc.), but rather the emotional experience that the respondent associated with it.

The first cluster, "suffering," and the second one, "emotional-sexual flattening," highlight a general experience of suffering due to social restrictions, immobility, and the absence of prospects for change in the long term that quarantine brought. The third cluster is the largest and most uneven: Some items within it are closer to others than they are with the rest of them, forming subgroups. However, these refer to the same phenomenon, the perception of sexual restrictions, focusing on different aspects. First, participants look at the purely emotional experiences arising from the possibility of expressing or not sexuality: The deprivation of a part considered important for one's life generated negative emotions, while, when sexual activity was retained, it represented a source of happiness and stress relief. Secondly, the change in the possibility of expressing sexuality pushed several participants to readjust their sexual repertoire and habits, attempting to reinvent themselves and find new kinds of sexual satisfaction. Thirdly, the experiences related to the possibility of expressing sexuality as a couple were reported by respondents who were searching for passion and tenderness, as well as desiring to see each other again. For other couples, sexual abstinence was experienced as a choice of loyalty to their partner, an additional sacrifice to the restrictions imposed.

In the second question about desire, arousal, and fantasies, "dream" is, for participants, the privileged space for sexual fantasies and where a "variety" of new sexual roles, orientation, and behaviors are experienced as an "opportunity for transgression." Despite the fact that approximately 47% of participants reported being followers of a religion (see also S1 Table), dreams and fantasies do not seem to be affected. Most religions (definitely Catholicism, Orthodoxy, and Buddhism) have doctrines that are quite restrictive on sexuality. They have pretty much told their followers that they should not do anything other than put penises in vaginas, and even that, ideally, should only take place within the confines of a heterosexual, monogamous marriage. Desires for any other sexual activities have been deemed unnatural, immoral, and unhealthy, discouraging one from acting on them with threats of divine retribution [47]. However, from our sample it emerges that the lockdown condition itself was not experienced by everyone as a reduction in sexual desire, fantasies, and arousal.

In the third question asking about masturbation, the first cluster consists of only one term, "privacy," a very influential dimension of sexual privacy and intimacy. If, in the first cluster, privacy expresses a holistic category of personal life, in the second one it takes shape in domestic spaces of life at the time of quarantine: "need for space." It encompasses stems that reveal as the lockdown as much as it has distanced people socially, so it has brought them exaggeratedly closer together, in a promiscuity of spaces and times that deprive them of the necessary interpersonal distance needed for an authentic shared and liberating sexual intimacy. Therefore, self-eroticism was confined to the only place perceived as safe (i.e., the bathroom and shower). Nonetheless, for other respondents, masturbation intensified and was seen as a means to relax and calm down, a daily appointment with oneself, as shown by the third cluster "escape from the room." In this cluster, stems are associated with the positive role of masturbation as stress relief.

For the fourth question about the use of sexual aids, the salient stems provide information about the variety of aids participants reported using, which were purchased and used as tools for coping with the lockdown. Participants said they had purchased vibrators, or accessed sexually explicit Internet sites (e.g., YouPorn.com, Pornhub.com), or resorted to reading short erotic stories or watching erotic films, as well as exchanging pictures and videos with their partner (sexting). These findings align with data provided by Pornhub [50] in a report on statistics regarding access to its website. Pornhub recorded that Italian user traffic increased significantly compared to the pre-COVID-19 average traffic in February 2020 (i.e., before the outbreak), with a drastic 57% increase on March 12, 2020, when the platform offered free access to the premium service.

The last question was on awareness about sexuality. The stems loading in the first cluster reveal that awareness of one's sexual experience has increased emotional literacy and the value of sexual intimacy about both oneself and one's partner, a fundamental good not to be lost or neglected. The testimonies collected from this group of tweets reveal that participants experienced this exceptional time not only as a tragedy, painful and deadly, but also as an opportunity for growth. Elements such as bonding, tenderness, emotions experienced with a partner, and playful moments such as foreplay all contributed to full sexual satisfaction. As early as 2017, in a much-cited literature review on the effects of social isolation and loneliness on health, Leig-Hunt and colleagues [51] found that a quality of relationships is crucial in the prevention of psychological distress due to social isolation, than quantity of them. The second cluster, "accepting sexuality," is rather uneven; it includes additional groupings within it, based on the greater proximity of certain items. However, it is possible to observe that these refer to different facets of the same dimension: accepting sexuality, both as an aspect of one's person and in its multiple meanings. The participants emphasized the complexity of sexuality, which can be understood not only as a physical relationship but also as part of a loving relationship, which focuses on gestures of intimacy, affection, and mutual attraction. Finally, some participants report reintegrating sexuality into their lives during the lockdown period, setting aside feelings of guilt and gaining an understanding that it can be an important dimension to devote time to, allowing themselves to be "sexually interesting."

During the review process of the present study, one of the reviewers pointed out to us a limitation regarding the wording of the Open-ended e-Questionnaire on Sexual Experience. Some of the five questions included more than one question, which might suggest a possible answer. For example, question 2: "How did you do with your sexual desire and arousal? Have you noticed a change in your erotic fantasies?": The second question might suggest that there was a change in sexual fantasies. This might have led respondents to not completely frank answers, according to the reviewer.

Another limitation of the study concerns the use of the Kinsey scale to identify sexual orientation. We used the (old) seven-item version which did not include to account for asexuality. This may have limited some respondents in finding appropriate identification regarding their sexual orientation.

## Conclusions

In the present study, we investigated, through a phenomenological analysis, the sexual experience of young Italian adults in the lockdown condition experienced during phase one of the Italian governmental measures for the containment of the COVID-19 pandemic.

Despite the difficulties related to isolation and physical distance from their partners, the lockdown period offered to many an opportunity for self-awareness and reflection on their sexual experience, allowing questions about their sexual needs and fantasies to surface to their consciousness and feelings. The understanding, the bond, the emotions, and the affection that were established with one's partner were found to be a fundamental asset, not to be lost, and contributed to full sexual satisfaction. At the same time, participants have rediscovered in the relationship with their partners the value, richness, and complexity of sexual intercourse, comprised of play, foreplay (petting), kissing, and gestures of tenderness. In this way, participants claimed to have overcome a forced sexual intercourse limited to penetration and orgasm attainment alone and to have regained the meaning of masturbation as a normal and healthy expression of their individual and couple sexuality.

From the initial conflict and confusion of the new existential condition due to physical and social isolation or over-proximity, Italian adults turned to creativity in response to the

pandemic tragedy, discovering new forms of intimacy, consisting of loving and erotic close-ness as well as care and acceptancy of their own sexual bodies. Sexuality has thus been rede-fined in terms of one's personal search for identity, remaining one of the fundamental ways in which a person expresses themselves in the world through the experience of pleasure even dur-ing a tiring and painful quarantine.

## Supporting information

**S1 Table. Frequency information of total wordlist extracted from the answers to open-ended questionnaire.** Frequency information about words extracted from answers to open-ended questions are reported. In the first column. word stems are shown. Frequencies (F(1–5)). total frequencies (F(total)). term frequencies (TF(1–5)). inverse document frequencies (IDF) and term-frequency-inverse document frequencies (TF-IDF(1–5)) related to the five context are reported for each term. The last row "Total" shows the measure of the length of each docu-ment, where the latter is considered as the total number of stems for each open-ended question multiplied by their frequency *(F(1–5))*.
(DOCX)

**S2 Table. Euclidean distance matrix between stems with higher TF-IDF in question 1.** Euclidean distance between the 31 terms with higher TF-IDF in the text corpus of answers to open-ended Question 1 are reported. The greater the Euclidean distance, the greater the dis-tance/dissimilarity between items.
(DOCX)

**S3 Table. Euclidean distance matrix between stems with higher TF-IDF in question 2.** Euclidean distance between the 7 terms with higher TF-IDF in the text corpus of answers to open-ended Question 2 are reported. The greater the Euclidean distance, the greater the dis-tance/dissimilarity between items.
(DOCX)

**S4 Table. Euclidean distance matrix between stems with higher TF-IDF in question 3.** Euclidean distance between the 10 terms with higher TF-IDF in the text corpus of answers to open-ended Question 3 are reported. The greater the Euclidean distance, the greater the dis-tance/dissimilarity between items.
(DOCX)

**S5 Table. Euclidean distance matrix between stems with higher TF-IDF in question 5.** Euclidean distance between the 16 terms with higher TF-IDF in the text corpus of answers to open-ended Question 5 are reported. The greater the Euclidean distance, the greater the dis-tance/dissimilarity between items.
(DOCX)

**S6 Table. Agglomeration Schedule for Complete Linkage of stems with higher TF-IDF in question 1.** Summary of cluster solutions. At each stage, the cases with the smallest Euclidean distance are combined; the coefficients indicating cluster heterogeneity change when a case is combined with the cluster. The solution before the largest gap in the coefficient indicates the best cluster solution (Stage 28).
(DOCX)

**S7 Table. Agglomeration schedule for complete linkage of stems with higher TF-IDF in question 2.** Summary of cluster solutions. At each stage, the cases with the smallest Euclidean distance are combined; the coefficients indicating cluster heterogeneity change when a case is

combined with the cluster. The solution before the largest gap in the coefficient indicates the best cluster solution (Stage 4).
(DOCX)

**S8 Table. Agglomeration schedule for complete linkage of stems with higher TF-IDF in question 3.** Summary of cluster solutions. At each stage, the cases with the smallest Euclidean distance are combined; the coefficients indicating cluster heterogeneity change when a case is combined with the cluster. The solution before the largest gap in the coefficient indicates the best cluster solution (Stage 7).
(DOCX)

**S9 Table. Agglomeration schedule for complete linkage of stems with higher TF-IDF in question 5.** Summary of cluster solutions. At each stage, the cases with the smallest Euclidean distance are combined; the coefficients indicating cluster heterogeneity change when a case is combined with the cluster. The solution before the largest gap in the coefficient indicates the best cluster solution (Stage 14).
(DOCX)

**S10 Table. Frequencies of participants' responses to sociodemographic questionnaire for binary gender (male/female) derived from responses to the question "sex as assigned at birth".**
(DOCX)

## Author Contributions

**Conceptualization:** Stefano Federici, Rosella De Leonibus, Anna Maria Acocella, Adriana Giammaria.

**Data curation:** Stefano Federici, Alessandro Lepri, Alessandra Castellani Mencarelli, Evel Zingone.

**Formal analysis:** Alessandro Lepri, Alessandra Castellani Mencarelli, Evel Zingone.

**Investigation:** Stefano Federici, Rosella De Leonibus, Anna Maria Acocella, Adriana Giammaria.

**Methodology:** Stefano Federici, Alessandro Lepri, Alessandra Castellani Mencarelli, Evel Zingone, Anna Maria Acocella, Adriana Giammaria.

**Supervision:** Stefano Federici, Rosella De Leonibus, Anna Maria Acocella.

**Writing – original draft:** Stefano Federici, Alessandro Lepri, Alessandra Castellani Mencarelli, Evel Zingone.

**Writing – review & editing:** Stefano Federici, Alessandra Castellani Mencarelli, Rosella De Leonibus.

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
