## [Decision Letter · Decision Letter 0]

14 Jan 2022

PONE-D-21-17508The sexual experience of Italian adults during the COVID-19 lockdownPLOS ONE

Dear Dr. Federici,

Thank you for submitting your manuscript to PLOS ONE. After careful consideration, we feel that it has merit but does not fully meet PLOS ONE’s publication criteria as it currently stands. Therefore, we invite you to submit a revised version of the manuscript that addresses the points raised during the review process. As you will see, the three reviewers were drastically split in their appraisals. I decided to take the middle line in hopes of providing you the opportunity to revise your paper. Once it is resubmitted, I will ask the three to give their opinion again, given this split-decision. Please submit your revised manuscript by Feb 28 2022 11:59PM. If you will need more time than this to complete your revisions, please reply to this message or contact the journal office at plosone@plos.org. Please include the following items when submitting your revised manuscript:A rebuttal letter that responds to each point raised by the academic editor and reviewer(s). You should upload this letter as a separate file labeled 'Response to Reviewers'.A marked-up copy of your manuscript that highlights changes made to the original version. You should upload this as a separate file labeled 'Revised Manuscript with Track Changes'.An unmarked version of your revised paper without tracked changes. You should upload this as a separate file labeled 'Manuscript'.

We look forward to receiving your revised manuscript.

Kind regards,

Peter Karl Jonason

Academic Editor

PLOS ONE

https://journals.plos.org/plosone/s/file?id=ba62/PLOSOne_formatting_sample_title_authors_affiliations.pdf”

Reviewers' comments:

Reviewer's Responses to Questions

**Comments to the Author**

1. Is the manuscript technically sound, and do the data support the conclusions?

Reviewer #1: Yes

Reviewer #2: No

Reviewer #3: Partly

2. Has the statistical analysis been performed appropriately and rigorously? 

Reviewer #1: Yes

Reviewer #2: No

Reviewer #3: No

3. Have the authors made all data underlying the findings in their manuscript fully available?

Reviewer #1: Yes

Reviewer #2: Yes

Reviewer #3: Yes

4. Is the manuscript presented in an intelligible fashion and written in standard English?

Reviewer #1: Yes

Reviewer #2: No

Reviewer #3: Yes

5. Review Comments to the Author

Reviewer #1: Thanks for sending the paper. I found it interesting and noble. However, while reading the abstract, I couldn't understand: TF-IDF

Result: I don't think, the first sentence should mention in such as manner.

Reviewer #2: The ms aimed to study sexuality during the lockdown in Italy, with a mixed quantitative/qualitative design study.

I think that the paper is not apt to be published in Plos One since it does not have the required standards.

The main problems are the quantitative analysis, which seems to be strange if not wrong at all, and the qualitative outcome, which appears to be invalidated by the nature of the five questions, more of which encompasses more questions and suggest the answer. I think that without a clear question without any suggestion of a possible answer data cannot be interpreted.

I suggest the author to write a purely qualitative study. I found it very interesting the considerations about disability, but I wonder if they really can be derived from the data analyzed in the way the authors did.

Major points:

Introduction section:

the importance of sexual life for general wellness is not introduced linearly. The authors speak about calls by international journals, but that cannot be a sign of the relevance of sexual health for general wellbeing during the pandemic. It is a consequence of that. The literature review, even if complete, is not delineated with clarity.

I even wonder about the word “Predictions” (line 149): it is better to use Hypotheses.

Method section:

Line 233: why did the authors use 56 (!) gender options drawn from Facebook? Facebook is not a scientific source and 56 are too many. I find it very confusing, especially in comparison with the Kinsey scale used for sexual orientation, which is old and does not encompass asexuality (sexual and romantic).

Line 263: I think the questionnaires were collected confidentially and not anonymously: no need to identify the subject since it was not a longitudinal study. Otherwise please specify what kind of code was required to participate and for what reason.

Some pieces of information about the method were put are in the Result section, making it difficult to follow the paper. Please move them into the Method section.

Line 329: I find it very strange to analyze a standardized questionnaire in this way: the authors should specify why they choose to use quintiles to analyze the first and the third scale of the SMQ.

Line: 372: the t-test cannot be used to investigate the frequency of answers: it should be used the chi-square test.

Furthermore, I don't see the necessity to analyze each item. A three scales questionnaire is better analyzed by a manova and post hoc.

Results:

Table 1: the table is the output of the SPSS: it encompasses p = .000 that is how the program writes the outcome but is wrong since probability cannot be = 0, since it is asymptotic to infinity. Other elements in this table are irrelevant: please eliminate them.

I don't understand many cluster names the authors used, without always defining them, such as:

Line 464: ‘“dream” indicate where fantasies were expressed’: dream does not indicate a place, nor the example in line 465 speaks about places;

“opportunity for transgression”: the example in line 475 does not speak of any transgression;

“privacy”: the example in line 488 speaks about autoeroticism…

“need for space”: the example in line 494 speaks of privacy, that was another cluster…

“need for intimacy”: the example inline 529 speaks about a deepest awareness of sexual orientation and of gender and sexual identities. It is not about intimacy…

“accepting sexuality”: the example in line 539 is not about accepting something, but about a deeper understanding of what sexuality is.

Discussion:

I find it very interesting the discussion about disability, but I don't think the data says what the authors claim. To reach such conclusions is required a different kind of analysis.

I find it very interesting the discussion about open ended questions: still, the data presented did not allow to drawn such conclusions.

Some points are difficult to follow since the English is not so clear: please make a professional revision.

Reviewer #3: In this paper, the authors addressed the critical issues about the effect on sexual health and behavior due to the lockdown period experienced by 465 Italian participants. Nowadays, the topic is of great interest as the Covid-19 emergency has forced people to stay in quarantine, impacting their physical, psychological, and, no doubt, sexual health. I think this work offers valuable insight into this critical issue.

To investigate the sexual health and behaviors of Italian adults during the lockdown period, authors used three main instruments:

1. A Sociodemographic and Behavioral e-Questionnaire During Quarantine;

2. The Sexual Modes Questionnaire (SMQ) – Nonbinary Form: a closed-ended self-report questionnaire (five-point Likert-type scale) to assess the interactions between cognition, emotion, and sexual response.

3. An Open-ended e-Questionnaire on Sexual Experience.

Here, participants were invited to respond to five open-ended questions with a maximum of 280 characters. In the analysis of this last instrument, the TF-IDF index was used to extract the weight of some salient words employed by respondents to answer the five open-ended questions.

I will focus precisely on this type of analysis as some points need to be clarified.

Please, find my review comments as an attachment file.

6. PLOS authors have the option to publish the peer review history of their article (what does this mean?). If published, this will include your full peer review and any attached files.

Reviewer #1: No

Reviewer #2: No

Reviewer #3: No

---

## [Author Response · Author response to Decision Letter 0]

16 Feb 2022

Reply to reviewers’ comments

Below we provide a point-by-point reply to the reviewers’ comments and suggestions.

• R1 = Reviewer 1 comment; RP1 = Reply to Reviewer 1 comment

• R2 = Reviewer 2 comment; RP2 = Reply to Reviewer 2 comment

• R3 = Reviewer 3 comment; RP3 = Reply to Reviewer 3 comment

Reviewer 1 

1. R1: “Thanks for sending the paper. I found it interesting and noble. However, while reading the abstract, I couldn’t understand: TF-IDF.”

• RP1: Thank you very much for your positive appreciation of our study. Regarding the abstract, we have modified the reference to TF-IDF as follows:

“In the qualitative analysis, the Frequency-Inverse Document Frequency (TF-IDF) index was computed to measure the salience of the word used by participants to respond to the open-ended five questions.”

2. R1: “Result: I don’t think, the first sentence should mention in such as manner.”

• RP1: While it is not unusual to find the data availability statement in Results, we have, however, addressed your comment by moving the sentence to the first paragraph of the Statistical Analyses subsection.

Reviewer 2

3. R2: “The ms aimed to study sexuality during the lockdown in Italy, with a mixed quantitative/qualitative design study. I think that the paper is not apt to be published in Plos One since it does not have the required standards. The main problems are the quantitative analysis, which seems to be strange if not wrong at all, and the qualitative outcome, which appears to be invalidated by the nature of the five questions, more of which encompasses more questions and suggest the answer. I think that without a clear question without any suggestion of a possible answer data cannot be interpreted. I suggest the author to write a purely qualitative study. I found it very interesting the considerations about disability, but I wonder if they really can be derived from the data analyzed in the way the authors did.”

• RP2: We want to thank you for your careful analysis of our manuscript, although we are sorry that you did not appreciate it. We hope that this new version of the manuscript, richly reworked based on the suggestions provided by you and the two other Reviewers, will receive a more favorable judgment from you. As you will read below, Reviewer #3, as an expert in statistics, raised no issue with the analysis techniques adopted on both quantitative and qualitative data. He or she, focusing exclusively on the qualitative analysis, required us to clarify our procedures. Neither of the two other Reviewers raised any questions about whether our data support our conclusions; nor did they require any changes in this regard. Of course, this does not mean that your doubts about the validity of our conclusions—for example, about disability—and the correctness of our data analysis are therefore unfounded. For these reasons, we will try to answer your comments point by point, within the limits of our possibilities.

4. R2: “Introduction section: the importance of sexual life for general wellness is not introduced linearly. The authors speak about calls by international journals, but that cannot be a sign of the relevance of sexual health for general wellbeing during the pandemic. It is a consequence of that. The literature review, even if complete, is not delineated with clarity.”

• RP2: We have never asserted that “the relevance of sexual health for general wellbeing during the pandemic” was due to the space that international journals gave to this topic. On the contrary, we agree with you that the fact that prestigious international journals reserved ample space on the topic of sexuality and COVID is evidence of the consequences of the pandemic for sexual health for general wellbeing. In terms of the clarity with which our literature review was laid out, we proceeded to submit the text to a professional English language proofreader whose changes you can check in the Introduction section in both the clean and tracked versions of the revised manuscript.

5. R2: “Introduction section: I even wonder about the word “Predictions” (line 149): it is better to use Hypotheses.”

• RP2: Thanks for the suggestion. We proceeded to replace the term “predictions” with “hypotheses”.

6. R2: “Method section: Line 233: why did the authors use 56 (!) gender options drawn from Facebook? Facebook is not a scientific source and 56 are too many. I find it very confusing, especially in comparison with the Kinsey scale used for sexual orientation, which is old and does not encompass asexuality (sexual and romantic).”

• RP2: We share your concern that too many choices about gender identity can be confusing to the respondent. In fact, in the research we are currently conducting, we have introduced the only six options proposed by Broussard and colleagues [Broussard, K. A., Warner, R. H., & Pope, A. R. D. (2018). Too many boxes, or not enough? Preferences for how we ask about gender in cisgender, LGB, and gender-diverse samples. Sex Roles, 78(9), 606–624. https://doi.org/10.1007/s11199-017-0823-2]. However, it must also be said that the Facebook list is no less “scientific” than Broussard et al.’s, who conducted the study to reduce the options to only six items precisely because they had found how widespread the use of the Facebook list was in scientific studies. Last but not least, Broussard et al.’s list cannot be considered more “scientific” than Facebook’s because it is drawn from a survey of respondents’ preferences and not the development of a standardized psychometric measure of gender identity.

With regard to the Kinsey scale, it is old but not for this reason obsolete. We are familiar with the Kinsey scale debate that animates the scientific debate over gender from Kinsey’s Heterosexual-Homosexual Rating Scale to Klein’s Sexual Orientation Grid to the more recent sexual configurations theory, while the U.S. Department of Health and Human Services survey data collection (https://aspe.hhs.gov/reports/guide-hhs-surveys-data-resources) efforts have focused on less complex measures of sexual orientation [Wolff, M., Wells, B., Ventura-DiPersia, C., Renson, A., & Grov, C. (2017). Measuring sexual orientation: A review and critique of U.S. data collection efforts and implications for health policy. The Journal of Sex Research, 54(4–5), 507–531. https://doi.org/10.1080/00224499.2016.1255872]. Far from being abandoned, the Kinsey scale is still widely used. It is also not true that the Kinsey scale does not contemplate asexuality. The recent version of the scale indeed introduces item X” “No socio-sexual contacts or reactions”.

7. R2: “Method section: Line 263: I think the questionnaires were collected confidentially and not anonymously: no need to identify the subject since it was not a longitudinal study. Otherwise please specify what kind of code was required to participate and for what reason.”

• RP2: We had to adhere to the rules of the Bioethics Committee of the University of Perugia (protocol no. 51854/2020), which mandated that all information be collected anonymously (not confidentially). This Bioethics Committee’s norm is based on Regulation (EU) 2016/679 of the European Parliament and of the Council of 27 April 2016 on the protection of natural persons with regard to the processing of personal data and on the free movement of such data, and repealing Directive 95/46/EC (General Data Protection Regulation). About the protocol code, according to your request, we have specified in the manuscript as it was generated as follows:

“Responses to the questionnaires were collected anonymously: participants were assigned an anonymous alphanumeric protocol code, automatically generated by the Qualtrics.xm platform, and used by researchers for data processing.”

8. R2: “Method section: Some pieces of information about the method were put are in the Result section, making it difficult to follow the paper. Please move them into the Method section.”

• RP2: Taking your comments and those of Reviewer #3, the Methods section has been extensively revised and made clearer and more understandable in its content. We hope this meets your requirements. If not, please tell us which parts you still suggest we should change or move.

9. R2: “Method section: Line 329: I find it very strange to analyze a standardized questionnaire in this way: the authors should specify why they choose to use quintiles to analyze the first and the third scale of the SMQ.”

• RP2: Quintiles are used to create cut-off points for a given population. Quintiles are representative of 20% of a given population. Therefore, the first quintile represents the lowest fifth of the data and final quintile represents the final or highest fifth of the data. The mean and standard deviation are useful to summarize a set of observations, but when the data have a skewed distribution, it is often preferable to quote the quintile instead. In the present study, we chose quintiles for two reasons: (i) this metric in statistical analysis allowed us to distribute the sample by balanced age groups which would not have been balanced using mean and standard deviation; (ii) both the first and third SMQ scales had a skewness ±3.29 [Mayers, A. (2013). Introduction to statistics and SPSS in psychology. Pearson. p. 53].

10. R2: “Method section: Line: 372: the t-test cannot be used to investigate the frequency of answers: it should be used the chi-square test.”

• RP2: Thank you so much for your comment, which highlighted a deficiency in our use of the word “frequency” in the manuscript, causing confusion in the reader and justifying your comment. In the sections Method and Results, by “frequency”, we were not referring to “statistical frequency” but to the frequency with which an “automatic thought” occurred. As explained in the subsection Materials and Apparatus, for each automatic thought (30 items for male and 33 items for female), the respondents had to indicate on a five-point Likert-type scale the frequency with which a thought or an image occurred (from 1 = “never” to 5 = “always”). In the manuscript, therefore, by “frequency”, we refer to the Likert value indicating the occurrence of an automatic thought. That said, and again apologizing for being unclear in the original text, it seems fair to us to use a t-test to analyze the effect that sex (independent variable) had on the values of frequencies of automatic thoughts. Therefore, we have modified the manuscript as follows.

Results, subsection Quantitative Analysis on SMQ scales:

“…t-tests and one-way ANOVAs were conducted to compare the effect of the variables collected through the sociodemographic questionnaire (independent variable) on the response values that participants attributed to the SMQ questionnaire scales (dependent variable).”

Results, subsection Sex Differences on SMQ scales:

“A t-test was computed to investigate the effect of the independent variable sex (male or female based on the answers to “sex as assigned at birth”) on the occurrence of automatic thoughts (AT scale, dependent variable), indicated by respondents through a five-point Likert-type scale anchored by 1 (never) to 5 (always). As displayed in Table 1, significant differences in values were found among 13 items of the first scale AT.”

Results, subsection Disability Effect on SMQ Scales

“One-way ANOVAs were conducted to compare the effect of the variables collected via the sociodemographic questionnaire (independent variable) on the occurrence of automatic thoughts (AT scale, dependent variable), indicated by respondents through a five-point Likert-type scale anchored by 1 (never) to 5 (always).”

Table 1 and Table 4 have been amended.

11. R2: “Furthermore, I don’t see the necessity to analyze each item. A three scales questionnaire is better analyzed by a manova and post hoc.”

• RP2: Because there is no true standardization of the Italian adaptation, the most correct use is to analyze the individual items in order to extrapolate precise information on the performance of each of them. Analyzing the three scales in a general way without paying attention to the individual items would not give us accurate information.

12. R2: “Results: Table 1: the table is the output of the SPSS: it encompasses p = .000 that is how the program writes the outcome but is wrong since probability cannot be = 0, since it is asymptotic to infinity. Other elements in this table are irrelevant: please eliminate them.”

• RP2: We have amended Table 1 as you suggested.

13. R2: “Results: I don’t understand many cluster names the authors used, without always defining them, such as: Line 464: ‘“dream” indicate where fantasies were expressed’: dream does not indicate a place, nor the example in line 465 speaks about places; “opportunity for transgression”: the example in line 475 does not speak of any transgression; “privacy”: the example in line 488 speaks about autoeroticism… “need for space”: the example in line 494 speaks of privacy, that was another cluster… “need for intimacy”: the example inline 529 speaks about a deepest awareness of sexual orientation and of gender and sexual identities. It is not about intimacy… “accepting sexuality”: the example in line 539 is not about accepting something, but about a deeper understanding of what sexuality is.”

• RP2: Thank you for your comment on cluster analysis. Although we actually defined all cluster names and tried to report the most meaningful statements among the responses given by participants, it is possible that some opacity may be perceived due primarily to two factors. Firstly, all the collected texts of the answers to the five questions introduced on Atlas.ti software were in the Italian language and translated for this paper. The translation process, which did not involve a process of adaptation to the Anglo-Saxon culture and language, may have conveyed a slightly different meaning of the statements reported from the original Italian, causing some confusion for Reviewer #2. To overcome all this, we have provided a more careful and judicious choice of sample respondents’ sentences, also using a more accurate translation into English. The second factor has to do with the numerosity of the answers underlying the cluster analysis and represented by us in only a few sample sentences. A total of 2,311 tweets were evaluated, and the choice of a few sample expressions reported in Results may not have fully captured the complexity of the answers. Anyway, accepting your requests, we have modified the text as follows.

About the “Second Question: Desire, Arousal, and Fantasies”

“The first cluster, ‘dream’, indicates a phenomenon through which fantasies were expressed. ‘Both desire and fantasies increased, expressed more during dreams (especially during afternoon rest). Upon awakening, arousal was strong and long-lasting. It happened more rarely before the lockdown’ (Woman, 32 years old).”

“The third cluster points out that, for some respondents, fantasies were experienced as an ‘opportunity for transgression’.”

“’I have discovered and also experienced new fantasies related to the BDSM world and in particular, submission’ (Woman, 24 years old).”

About the “Third Question: Masturbation”

“The first cluster consists of only one very salient stem, ‘privacy’. ‘Unfortunately, my moments of privacy have been abruptly reduced and I haven’t had the opportunity to masturbate with the same tranquility and frequency as before’ (Woman, 27 years old).”

“The second cluster refers to the reduction of space and time for solitude and self-eroticism. ‘Due to lack of space, privacy, or discomfort, desire has often been experienced with frustration, as an impediment’ (Man, 28 years old).”

About the “Fifth Question: What has been Understood about One’s Sexuality”

“The second cluster, ‘accepting sexuality’, refers to different facets of the same dimension: accepting sexuality both as an aspect of one’s person and in its multiple meanings. ‘During this period of quarantine with respect to sexuality, I learned to indulge my moods by sharing them with my partner. I felt no shame for the lack of desire, the scarce excitement’ (Woman, 29 years old).”

14. R2: “Discussion: I find it very interesting the discussion about disability, but I don’t think the data says what the authors claim. To reach such conclusions is required a different kind of analysis. I find it very interesting the discussion about open ended questions: still, the data presented did not allow to drawn such conclusions.”

• RP2: Thank you for enjoying our discussion and interpretation of data regarding disability. We regret that you do not fully agree with the soundness of our analyses based on the data collected. This does not seem to be the case to us, and while a qualitative analysis certainly leaves a great deal of room for interpretation, we have deferred to the expressions used by the respondents and the data collected. We are willing to accept any suggestions you may have for improving our interpretation of the data, and we ask you kindly to tell us more precisely which statements you feel are not well-founded. Since we could not use the comments of the other two reviewers, who appreciated the discussions and conclusions to the study, we could not guess which points you wanted us to modify. 

15. R2: “Some points are difficult to follow since the English is not so clear: please make a professional revision.”

• RP2: We submitted the original and, according to your request, the revised manuscript to Proof-reading-service.com (reference no. = 202104-24204216 and no. = 202202-11125958).

Reviewer 3 

16. R3: “After that participant responded to the five open-ended questions with a maximum of 280 characters, the authors extracted some relevant words to the questions asked and the study. Subsequentially, they isolated a “stoplist,”: a set of empty words (articles, pronouns, conjunctions, etc.). Thus, a total number of 1.313 words (not belonging to this stoplist) were then considered for further analysis. On page 19, line 278, the authors wrote: “to facilitate the analysis, the 1,313 words not belonging to the stoplist were associated with each other based on the root of each word (i.e., the part not subject to variation, which contains the fundamental meaning of the word), obtaining a final list of 607 roots S1 Table.” Confirming this, the authors reported in the S1 Table description: “Roots (N= 607).” Despite that, in the S1 Table, I counted a little over 300 words, not 607. That being said, it’s not clear for me if this is a simple typing error or if further operations to the original list of 607 roots were carried out and not reported in the main text.”

• RP3: We are really grateful for all your thoughtful and insightful comments that allowed us to enrich our manuscript and make it more technically sound. With regard to your first comment, we regret to note that we made a typing error in Table S1, which we have amended. Now, all 607 extracted stems are reported.

17. R3: “In addition, it is essential to point out that the lemmatization procedure from which the authors then derive what they have called ‘root words’ is a standard procedure that has not been mentioned. Therefore, I recommend the authors provide the correct mention of the methodology adopted. It is essential, in my opinion, to clarify this point since the total number of root words is indispensable for estimating the TF-IDF index.”

• RP3: As you requested, we provided the methodology adopted—i.e., the stemming procedure instead of lemmatization—as follows:

“Then, to facilitate the analysis, the stemming procedure was used to reduce inflected (or derived) words to their word stem (i.e., the main part of a word that stays the same when endings are added to it). Whole words (i.e., not the stem) were maintained when present only once in the list. The 1,313 words not belonging to the stoplist were associated with each other based on the stem of each word, obtaining a final list of 607 stems (S1 Table).”

Also, we have replaced the word ‘root’ referring with stem, following the Oxford English grammar. In fact, while in Italian grammar “radice � root” is used both to refer to the invariant part of a word and to the part of a word from which other words are formed, we checked that in English grammar the two meanings are expressed with two different words.

18. R3: “After this first operation, the authors used the TF-IDF index to calculate the weight of each root within what they call the “5 contexts,” i.e., the five questions that were asked to the participants in the questionnaire defined “Open-ended e-Questionnaire on Sexual Experience.” From my point of view, the computation of the TF-IDF index is not well explained in the main text; I suppose that an inexperienced reader who is not familiar with the formula will not fully understand its meaning. I would suggest the authors add the TF-IDF index formula and clarify what is referred to as TF and what is referred to as IDF. For example: “… given a collection C of documents d, the TF-IDF value for each term t in a document d ∈ C is calculated as: 

TF-IDF(t,d,C) = TF (d,t) * IDF (t,C),

where TF (d, t) indicates the number of times a target term t appears in document d, and IDF is equal to log(N/n), where N indicates the number of documents in C and n is the number of documents where t is used. In this study case, the TF part of the formula, the collection C is the total number of documents (C =…), the term t corresponds to each of the roots (e.g., “abbracc*”), and the document d is equal to each one of the five questions belonging to the collection C. Concerning the IDF formula instead, N is equal to etc...”. If this is not clarified, I think it is not easy to understand how the TF-IDF was calculated.

• RP3: Thank you very much for your suggestions and support. We have added the TF-IDF formula and clarified point by point the computational process as follows:

“The Term Frequency-Inverse Document Frequency [TF-IDF; 31] weight function was applied to the list of stems. TF-IDF allowed us to measure the relevance that a word takes in its context of use. There are five contexts of use in this study, each of which collects all tweets in response to each of the five questions (sexual activity; desire, arousal, and fantasies; masturbation; use of sexual aids; awareness of sexuality). This measure of relevance is from the computation of inverse correlation between the frequency of a word stem among the tweets provided in response to each question and the frequency of the same stem among all 2,315 tweets gathered. Given a collection C of documents d, the TF-IDF value for each term t in a document d ∈ C is calculated as:

TF-IDF(t,d,C) = TF (d,t) * IDF (t,C),

where TF (d, t) indicates the number of times a target term t appears in document d, and IDF is equal to log(N/n), where N indicates the number of documents in C and n the number of documents where t is used. In this study case, the TF part of the formula, the collection C is the total number of documents (C = 5), the term t corresponds to each of the roots (e.g., “abbracci*”), and the document d is equal to each one of the five questions belonging to the collection C. To avoid favoring longer documents, the TF has been divided by the length of the document itself, where the latter is considered as the total number of stems for each open-ended question. Concerning the IDF formula instead, N is equal to the number of open-ended questions (N = 5) and n is equal to the number of questions where the stem appears. The salience of a stem (relative weight or TF-IDF) was considered higher the more its frequency in a specific context of use was inversely proportional to its frequency within the total number of 607 stems (i.e., the final list computed; S1 Table).”

19. R3: “Based only on what the authors reported, there are some points in the calculation of the TF-IDF index that are not clear to me: Starting from the TF, if I correctly understood, in your case, TF corresponds to the number of times a root t appears in document d, i.e., in each one of the five questions taken separately. For example, as reported in the S1 Table, if the root “abbracci*” frequency is equal to 2 in question number 1, it means that (taking the answers of all participants) this root was written two times in the participants text answers. Therefore, in this case, since no other operation is described in the manuscript, the TF is expected to be a whole number, as it is a frequency index. Nevertheless, in table S1, the TF corresponds to the number 0.00037. However, given that the TF should be divided by the length of the document itself (to avoid favoring longer documents), does the TF correspond to a decimal number because the frequency (e.g., F=2) has been divided by the length of the document text? I can guess from line 289 (“its frequency in a specific context of use”) that it may have been divided by the total number of words in the document. Is this correct? If so, however, the length of each document is not clear since the authors extracted some words (e.g., the ones in the “stoplist”). Could the authors please explain the TF calculation they have made?”

• RP3: Regarding the TF calculation, we divided the frequency of a stem in each context of use (e.g., how many times the stem “abbracci*” appears in all the tweets in response to the open-ended question 1, F = 2) by the length of the document itself, where the latter is considered as the total number of stems for each open-ended question. For instance, with regard to “abbracci*”, the length of the document is the occurrence of the stems in the corpus of tweets in response to question 1, that is equal to 2,421.

20. R3: “Turning to the calculation of the IDF, again, in my opinion, the authors did not provide enough information. The IDF is calculated as the log(N/n). Since N indicates the total number of documents in C and n is the number of documents where t is used, could the authors specify what they have considered N and what they have considered n? From my point of view, in the case of the root “abbracci*,” n should be equal to 3 since this term appears in 3 of the five documents (questions 1, 2, and 5). While it is not clear which value authors used to N. Following their reasoning, it should be 5, as the total number of the questions is five; however, if we apply these values, the IDF results are not the same reported in the Table S1. To summarise the above, the procedure and methodology of useful words extraction and lemmatization are not well described and make way for interpretation by the reader. Moreover, I also recommend the authors provide the correct mention of the methodology adopted. Furthermore, both TF and IDF calculations are standard procedures that, in my opinion, must be well-described step by step. As presented now, it is unclear which components of the text were used to calculate the TF-IDF, thus not allowing the reproducibility of the results.”

• RP3: We regret that the procedure and methodology of the useful words ‘extraction’ and ‘lemmatization’ were not well described. We hope that the extensive revision made to the text, also based on your valuable suggestions, will make understanding the methodology clearer to the reader. About the calculation of TF-IDF, the formula log(N/n) was used, where N indicates the total number of contexts of use (N = 5) and n indicates the number of contexts of use in which the stem appears. Since the formula does not specify the base of the logarithm, we calculated the natural logarithm. It is possible that with a different logarithm base the value would be different, but the overall result of the analysis should not change.

21. R3: “One the last clarification on TF-IDF: in line 410, authors wrote: “After applying the TF-IDF [31] weight function, where “DF = document frequency” was considered the occurrence of a root in each of the five open-ended questions.” Do the authors refer to the TF?”

• RP3: Thank you for pointing it out. We referred to DF (document frequency) as the total number of occurrences of the stem in the documents of the whole collection. We thought it could explain in a more detailed way how the TF-IDF formula works, since Wikipedia also reports that: “a high weight in tf–idf is reached by a high term frequency (in the given document) and a low document frequency of the term in the whole collection of documents; the weights hence tend to filter out common terms”. Probably, inserted here, this statement is redundant and misleading for the reader. Therefore, we have eliminated “DF = document frequency was considered the occurrence of a root in each of the five open-ended questions.”

22. R3: “Moreover, in two parts of the manuscript (lines 906, 907), the ‘TF’ is translated as text frequency instead of term frequency.”

• RP3: We regret our typo, which we have corrected.

23. R3: “In the online Excel table with the raw data provided by the authors, the last column collects some emails from participants in which their first and last names are visible. I strongly recommend that the authors make the data anonymous so that participants’ identities cannot be traced.”

• RP3: We removed the previous file in which the emails were visible, and we created a new one without the last column. Please see: Federici, S., Lepri, A., Castellani Mencarelli, A., Zingone, E., De Leonibus, R., Acocella, A. M., & Giammaria, A. (2022). Raw data about the sexual experience of Italian adults during the COVID-19 lockdown. Researchgate.net. https://doi.org/10.13140/RG.2.2.13355.36640

24. R3: “From a practical point of view, the English language is clear and of sufficient quality to be understood. I remind authors to add the page number at the bottom of the pages.”

• RP3: We have entered the page number as you requested.

25. R3: “The tables and figures are well done, although I would make Table S10 in the same format as the others so that they are all similar.”

• RP3: We reformatted S10 Table according to your request.

---

## [Decision Letter · Decision Letter 1]

24 Mar 2022

PONE-D-21-17508R1The sexual experience of Italian adults during the COVID-19 lockdownPLOS ONE

Dear Dr. Federici,

Thank you for submitting your manuscript to PLOS ONE. After careful consideration, we feel that it has merit but does not fully meet PLOS ONE’s publication criteria as it currently stands. Therefore, we invite you to submit a revised version of the manuscript that addresses the points raised during the review process. Please submit your revised manuscript by May 08 2022 11:59PM. If you will need more time than this to complete your revisions, please reply to this message or contact the journal office at plosone@plos.org. Please include the following items when submitting your revised manuscript:A rebuttal letter that responds to each point raised by the academic editor and reviewer(s). You should upload this letter as a separate file labeled 'Response to Reviewers'.A marked-up copy of your manuscript that highlights changes made to the original version. You should upload this as a separate file labeled 'Revised Manuscript with Track Changes'.An unmarked version of your revised paper without tracked changes. You should upload this as a separate file labeled 'Manuscript'.

We look forward to receiving your revised manuscript.

Kind regards,

Peter Karl Jonason

Academic Editor

PLOS ONE

Reviewers' comments:

Reviewer's Responses to Questions

**Comments to the Author**

1. If the authors have adequately addressed your comments raised in a previous round of review and you feel that this manuscript is now acceptable for publication, you may indicate that here to bypass the “Comments to the Author” section, enter your conflict of interest statement in the “Confidential to Editor” section, and submit your "Accept" recommendation.

Reviewer #2: (No Response)

Reviewer #3: (No Response)

2. Is the manuscript technically sound, and do the data support the conclusions?

Reviewer #2: Partly

Reviewer #3: Yes

3. Has the statistical analysis been performed appropriately and rigorously? 

Reviewer #2: No

Reviewer #3: Yes

4. Have the authors made all data underlying the findings in their manuscript fully available?

Reviewer #2: Yes

Reviewer #3: Yes

5. Is the manuscript presented in an intelligible fashion and written in standard English?

Reviewer #2: (No Response)

Reviewer #3: Yes

6. Review Comments to the Author

Reviewer #2: I will quote my previous remarks for an easier understanding.

3R2: My major concerns remain the nature of the five questions, many of which encompassed more than one question and suggest a possible answer. For example question 2: “How did you do with your sexual desire and arousal? Have you noticed a change in your erotic fantasies?”: the second question suggests that there was a change in sexual fantasies; same for question 3; question 4 asks two different questions while question 5 three different ones.

The authors should make it clear in the Discussion section that this is a limitation of their study that may have led to not completely frank answers.

4R2: I found no tracks of any correction in the Introduction section, but for the change of “Predictions” into “hypotheses”. So my doubts are just the same as before. Please change the sentence about international publications: as it stands now it is misleading.

6R2: I find your answer not appropriate: Sex assigned at birth is not relevant in a psychological study: you should rerun your analysis considering gender identity, and limiting them to persons that identify themself as male or female since the other groups were too little and cannot be encompassed in the sample unless losing data clarity.

Furthermore, it is not important how other people or even you are carrying out other research, in this one you used the old Kinsey scale, quoting it, and that does not encompass asexuality. Please quote that in the study limitation.

9.R2: you just deleted that part from the paper. I know what quintiles are and their use: I don’t need a lesson about that. If the data are not normally distributed quintiles are not appropriate for determining a cut-off.

11. R2: I still think that t-test and Anova on single items are not appropriate tests. Furthermore, if this is allowed (but a statistician must answer to that) you should use the Bonferroni correction for the p-value: .05 x items N.

13.R2: I appreciate the changes

14.R2: In lines 617-619 you made an interesting hypothesis. Nevertheless, I find it very difficult to follow the discussion since there are no means reported in the paper and I don’t know the direction of the significant difference: if the mean difference is positive it means that persons with disabilities had a higher mean than persons without disabilities? I would appreciate it if you can put some examples from your data in this part of the Discussion section that emphasized this aspect.

Reviewer #3: I have read the authors' responses to my comments.

The authors fixed what I had pointed out and clarified some minor points I had asked them to justify. Above all, they removed the participants' emails from the Excel file containing the online research data to guarantee the anonymity of the participants.

Turning to the major points, the authors correctly updated the table by adding the missing elements from the list of 607 words they had indicated in the text. They also updated the TF-IDF values compared to the previous table uploaded.

The authors also provided more detailed information about the stemming process regarding the adopted procedure:

"Then, to facilitate the analysis, the stemming procedure was used to reduce inflected (or derived) words to their word stem (i.e., the main part of a word that stays the same when endings are added to it). Whole words (i.e., not the stem) were maintained when present only once in the list. The 1,313 words not belonging to the stoplist were associated with each other based on the stem of each word, obtaining a final list of 607 stems (S1 Table)."

However, it is still unclear if it is a process they did by a Self-developed software. In this case, I suggest the authors indicate the software used and the source.

Regarding the TF-IDF formula, the authors have specified which elements are used for the computation and clarified point by point the computational process. I believe that it is now more understandable, even for a non-expert reader, how the TF-IDF has been calculated, thus making the reproducibility of the results possible as well.

7. PLOS authors have the option to publish the peer review history of their article (what does this mean?). If published, this will include your full peer review and any attached files.

Reviewer #2: No

Reviewer #3: No

---

## [Author Response · Author response to Decision Letter 1]

20 Apr 2022

Response to reviewers’ comments

Below we provide a point-by-point response to the reviewers’ comments and suggestions.

• R2 = Reviewer 2 comment; RP2 = Response to Reviewer 2 comment

• R3 = Reviewer 3 comment; RP3 = Response to Reviewer 3 comment

Reviewer 2

1. R2: “I will quote my previous remarks for an easier understanding. 3R2: My major concerns remain the nature of the five questions, many of which encompassed more than one question and suggest a possible answer. For example question 2: “How did you do with your sexual desire and arousal? Have you noticed a change in your erotic fantasies?”: the second question suggests that there was a change in sexual fantasies; same for question 3; question 4 asks two different questions while question 5 three different ones. The authors should make it clear in the Discussion section that this is a limitation of their study that may have led to not completely frank answers.”

• RP2: According to your request, we have added your criticism in the Discussion section as follows:

“During the review process of the present study, one of the reviewers pointed out to us a limitation regarding the wording of the Open-ended e-Questionnaire on Sexual Experience. Some of the five questions included more than one question, which might suggest a possible answer. For example, question 2: “How did you do with your sexual desire and arousal? Have you noticed a change in your erotic fantasies?”: The second question might suggest that there was a change in sexual fantasies. This might have led respondents to not completely frank answers, according to the reviewer.”

2. R2: “4R2: I found no tracks of any correction in the Introduction section, but for the change of “Predictions” into “hypotheses”. So my doubts are just the same as before. Please change the sentence about international publications: as it stands now it is misleading.”

• RP2: It is not our intention to make a trivial facade change, a simple maquillage just to meet your requirements. We are sincerely sorry if our correction gave you that impression. We have consulted with other colleagues and searched the relevant scientific literature for what exactly you are asking us to change. But your request continues to be unclear to us.

Anyway, we tried to make some changes to address your comment. We are aware that these changes are minimal. But it is not our intention in any way to minimize your request. The current changes and the original text are based on the following methodological assumptions:

(i) Our study design does not allow us to control for all intervening variables and establish or hypothesize a causal direction between independent and dependent variables;

(ii) Our study design hypothesizes a positive directional relationship between the variables/factors;

(iii) Our hypotheses are uttered in declarative form (declarative hypothesis vs. null form) because we make a positive statement about the outcome of the study.

Having said that, we made the following changes. Before the subsection began, we included a sentence as follows:

“We next present our expectations about the association between the lockdown condition and sexual health and behaviors of Italian adults.”

We titled the subsection “Expectations” as suggested in scientific studies such as https://doi.org/10.1509/jm.15.0340.

We changed the first sentence of the subsection as follows:

“Our expectations assume that the lockdown situation experienced in the first months of the pandemic may have altered various aspects of Italians’ sexuality, due to the changes in daily life imposed by the current COVID-19 outbreak.”

3. R2: “6R2: I find your answer not appropriate: Sex assigned at birth is not relevant in a psychological study: you should rerun your analysis considering gender identity, and limiting them to persons that identify themself as male or female since the other groups were too little and cannot be encompassed in the sample unless losing data clarity.

• RP2: We would like to sincerely thank you for your careful review of our manuscript and for all the time you devoted to it. It has always been our intention to take your comments seriously, welcoming them as an opportunity to improve our study. However, we were stunned by your statement “Sex assigned at birth is not relevant in a psychological study.” One can certainly believe that the earth is flat, but it is not up to us to give justification as to why it is round. Please provide us with references from the scientific community stating that it is unjustified in psychological studies to split a sample by sex. If we have missed statements and studies from the American Psychological Association, American Psychiatric Association, or even PlosOne’s policy we are willing to reconsider our data analyses. Otherwise, we like every other study published in international scientific Journals, we do not have to justify why we divided the sample by sex, while we should justify why we should divide it by gender.

4. R2: “6R2: Furthermore, it is not important how other people or even you are carrying out other research, in this one you used the old Kinsey scale, quoting it, and that does not encompass asexuality. Please quote that in the study limitation”

• RP2: We have added this limitation at the end of the Discussion as follows:

“Another limitation of the study concerns the use of the Kinsey scale to identify sexual orientation. We used the (old) seven-item version which did not include to account for asexuality. This may have limited some respondents in finding appropriate identification regarding their sexual orientation.”

5. R2: “9R2: you just deleted that part from the paper. I know what quintiles are and their use: I don’t need a lesson about that. If the data are not normally distributed quintiles are not appropriate for determining a cut-off.”

• RP2: It was not our intention to teach you anything or rant by giving you a lesson on quantiles. If we have given you this impression, and we apologize, it is only because we are trying to justify to you our methodological choices. According to our scientific and statistical knowledge, we disagree with your claim that “If the data are not normally distributed quintiles are not appropriate for determining a cut-off.” Please give us references to scientific publications of statistics that support your comment. According to our knowledge of statistics and the most famous scientific manuals and Journal articles, we believe that we have used a correct data analysis. Now below, we do not want to teach you a lesson, but wish to base our methodological choices on scientific evidence. For example, Kenney and Keeprin (1966) – cited on Google Scholar almost 2000 times and by some authors who themselves have more than 8000 citations – explain that quantiles, and specifically quartiles, are used not only to describe but even to calculate a skewness, in the same way as the more famous Pearson’s Skewness does. This formula is called Bowley skewness or Galton skewness. As more recently mentioned by Jones et al. (2011), “quantile-based measures of kurtosis and their interaction with skewness-inducing transformations, identifying classes of transformations that leave kurtosis measures invariant.” (p. 89). This makes us argue not only that using quartiles a skewness is a routine procedure, but also that it has important advantages for our study.

6. R2: “11R2: I still think that t-test and Anova on single items are not appropriate tests. Furthermore, if this is allowed (but a statistician must answer to that) you should use the Bonferroni correction for the p-value: .05 x items N.”

• RP2: A statistician (Reviewer #3) has already declared, as you can check on Editorial Manager, that the manuscript is technically sound, and the statistical analysis has been performed appropriately and rigorously.

However, we reiterate what we have already explained in the previous “Response to reviewers’ comments”. We have no normative data for SMQ. The original English version of SMQ was composed of two forms: the male form with 30 items divided into 5 factors (failure anticipation thoughts, erection concerns thoughts, age- and body-related thoughts, negative thoughts toward sex and lack of erotic thoughts) and the female form with 33 items divided into 6 factors (sexual abuse thoughts, failure and disengagement thoughts, partner’s lack of erotic thoughts) and the female version with 33 items divided into 6 factors (sexual abuse thoughts, failure and disengagement thoughts, partner’s lack of affection, sexual passivity, lack of erotic thoughts and low self-body image thoughts). For the present study, we created a new “non-binary version” by combining the two versions that not only have a different number of items but also different factors. Understanding how each item behaves is extremely important to us since we do not have validated data on the factorial structure on this new Italian version.

About the Bonferroni correction, we use it by default how you can see from our output. We apologize for not specifying it in the article. We have added ita the line 357 of the main text as follows: “Where appropriate, Bonferroni’s corrections have been applied.”

7. R2: “13R2: I appreciate the changes.”

• RP2: Thank you for your appreciation.

8. R2: “14R2: In lines 617-619 you made an interesting hypothesis. Nevertheless, I find it very difficult to follow the discussion since there are no means reported in the paper and I don’t know the direction of the significant difference: if the mean difference is positive it means that persons with disabilities had a higher mean than persons without disabilities? I would appreciate it if you can put some examples from your data in this part of the Discussion section that emphasized this aspect.”

• RP2: We thank you very much for this comment that give as the chance to improve the quality of our study by making it clearer. We have amended Tables 2 and 3 by adding a column with “Mean Differences” between participants with and without disability. In addition, in the Discussion section we have included references to those tables in parentheses when discussing the effect of disability on sexual behavior (lines 584-585).

Reviewer 3 

9. R3: “I have read the authors’ responses to my comments. The authors fixed what I had pointed out and clarified some minor points I had asked them to justify. Above all, they removed the participants’ emails from the Excel file containing the online research data to guarantee the anonymity of the participants. Turning to the major points, the authors correctly updated the table by adding the missing elements from the list of 607 words they had indicated in the text. They also updated the TF-IDF values compared to the previous table uploaded. The authors also provided more detailed information about the stemming process regarding the adopted procedure:

“Then, to facilitate the analysis, the stemming procedure was used to reduce inflected (or derived) words to their word stem (i.e., the main part of a word that stays the same when endings are added to it). Whole words (i.e., not the stem) were maintained when present only once in the list. The 1,313 words not belonging to the stoplist were associated with each other based on the stem of each word, obtaining a final list of 607 stems (S1 Table).”

However, it is still unclear if it is a process they did by a Self-developed software. In this case, I suggest the authors indicate the software used and the source.

• RP3: Thank you for your comment. We extracted the stems manually. The main has been modified as follows: 

“Then, to facilitate the analysis, the stemming procedure (manually conducted and not supported by software) was used to reduce inflected (or derived) words to their word stem (i.e., the main part of a word that stays the same when endings are added to it).”

10. R3: “Regarding the TF-IDF formula, the authors have specified which elements are used for the computation and clarified point by point the computational process. I believe that it is now more understandable, even for a non-expert reader, how the TF-IDF has been calculated, thus making the reproducibility of the results possible as well.”

• RP3: We are happy to hear that the manuscript is now more understandable and the experiment reproducible thanks also to your comments.

References

Jones, M. C., Rosco, J. F., & Pewsey, A. (2011). Skewness-invariant measures of kurtosis. The American Statistician, 65(2), 89–95. https://doi.org/10.1198/tast.2011.10194

Kenney, J. F., & Keeping, E. S. (1966). Mathematics of statistics. Van Nostrand Company.

---

## [Editor Report · Decision Letter 2]

22 Apr 2022

The sexual experience of Italian adults during the COVID-19 lockdown

PONE-D-21-17508R2

Dear Dr. Federici,

We’re pleased to inform you that your manuscript has been judged scientifically suitable for publication and will be formally accepted for publication once it meets all outstanding technical requirements.

Kind regards,

Peter Karl Jonason

Academic Editor

PLOS ONE
---

## [Editor Report · Acceptance letter]

26 Apr 2022

PONE-D-21-17508R2 

The sexual experience of Italian adults during the COVID-19 lockdown 

Dear Dr. Federici:

I'm pleased to inform you that your manuscript has been deemed suitable for publication in PLOS ONE. Congratulations! Your manuscript is now with our production department. 

Kind regards, 

on behalf of

Dr. Peter Karl Jonason 

Academic Editor

PLOS ONE